# PRL3 induces polyploid giant cancer cells eliminated by PRL3-zumab to reduce tumor relapse

Min Thura [1,6], Zu Ye[2,6], Abdul Qader Al-Aidaroos[1], Qiancheng Xiong [3], Jun Yi Ong[1], Abhishek Gupta[1], Jie Li [1], Ke Guo[4], Koon Hwee Ang[1] & Qi Zeng [1,5✉]

PRL3, a unique oncotarget, is specifically overexpressed in 80.6% of cancers. In 2003, we reported that PRL3 promotes cell migration, invasion, and metastasis. Herein, firstly, we show that PRL3 induces Polyploid Giant Cancer Cells (PGCCs) formation. PGCCs constitute stem cell-like pools to facilitate cell survival, chemo-resistance, and tumor relapse. The correlations between PRL3 overexpression and PGCCs attributes raised possibilities that PRL3 could be involved in PGCCs formation. Secondly, we show that PRL3+ PGCCs co-express the embryonic stem cell markers SOX2 and OCT4 and arise mainly due to incomplete cytokinesis despite extensive DNA damage. Thirdly, we reveal that PRL3+ PGCCs tolerate prolonged chemotherapy-induced genotoxic stress via suppression of the pro-apoptotic ATM DNA damage-signaling pathway. Fourthly, we demonstrated PRL3-zumab, a First-in-Class humanized antibody drug against PRL3 oncotarget, could reduce tumor relapse in 'tumor removal' animal model. Finally, we confirmed that PGCCs were enriched in relapse tumors versus primary tumors. PRL3-zumab has been approved for Phase 2 clinical trials in Singapore, US, and China to block all solid tumors. This study further showed PRL3-zumab could potentially serve an 'Adjuvant Immunotherapy' after tumor removal surgery to eliminate PRL3+ PGCC stem-like cells, preventing metastasis and relapse.

[1] Institute of Molecular and Cell Biology, Agency for Science, Technology and Research, Singapore, Singapore. [2] MD Anderson Cancer Centre, The University of Texas, Houston, TX, USA. [3] Department of Cell Biology, Yale School of Medicine, New Haven, CT, USA. [4] Genome Institute of Singapore, Agency for Science, Technology and Research, Singapore, Singapore. [5] Department of Biochemistry, Yong Loo Lin School of Medicine, National University of Singapore, Singapore, Singapore. [6] These authors contributed equally: Min Thura, Zu Ye. ✉email: mcbzengq@imcb.a-star.edu.sg

Phosphatase of regenerating liver 3 (PRL3) that we identified in 1998[1] belongs to a unique family of C-terminal pre-nylated phosphatases within the protein tyrosine phosphatase superfamily[2]. In 2001, the Vogelstein group showed that *PRL3* mRNA was overexpressed in metastatic liver lesions compared to corresponding primary colorectal tumors or normal colon epithelium[3]. We demonstrated that PRL3 protein is highly expressed in an average of 80.6% of tumors across 11 common cancer types[4]. PRL3 has been previously demonstrated to drive cell migration, tumor progression, metastasis, and survival[5–9], and targeted deletion of PRL3 was shown to suppress malignant transformation[10]. Interestingly, PRL3 induces a stem cell-like transcriptional program in acute myeloid leukemia (AML) by upregulating LIN28B, a stem cell reprogramming factor[11]. More recently, PRL3 expression was induced upon genotoxic stress to promote cancer growth and survival[12].

Polyploid giant cancer cells (PGCCs) are a unique subpopulation of grossly hyperdiploid cancer cells that differ from the bulk of dominantly aneuploid tumor cells in their morphology, tumorigenic ability, radio-resistance, and chemo-resistance. The vast majority of cancers are aneuploid, with around 90% of solid tumors and 75% of hematopoietic cancers having abnormal chromosome numbers[13]. Importantly, aneuploidy has been shown to precede transformation in a variety of cancers, suggesting that it is involved in tumor initiation and progression[14–16]. Studies with human prostate cancer cell lines demonstrated that PGCCs are more aggressive, metastatic, and highly resistant to common chemotherapeutic drugs including cisplatin, doxorubicin, and 5-fluorouracil as compared to mono-nucleated cancer cells[17]. PGCCs constitute stem cell-like pools facilitating cancer cell survival, therapy resistance, and tumor relapse. PGCCs express various embryonic stem cell markers[18,19]. Recently, PGCCs was reported to provide a survival advantage to hypoxic tumor cells by generating erythroid cells and inducing neo-angiogenesis via vasculogenic mimicry[20].

These striking similarities between PRL3 overexpression and PGCCs attributes raised the intriguing possibility that PRL3 could be involved in PGCC formation. We show here that PRL3 overexpression induces the formation of PGCCs, which displayed marked resistance to cisplatin chemotherapy by inhibiting pro-apoptotic DNA damage signaling pathways, thus enhancing cancer cell survival.

Previously, we have developed PRL3-zumab against PRL3 oncoprotein to block multiple PRL3 positive tumors in several animal models[4,21]. After more than 10 years of unconventional cancer immunotherapy in mice, in 2017, PRL3-zumab was administered First in Man at the National University Hospital (NUH) Singapore. A Phase I clinical trial has been successfully completed to show a safety profile of PRL3-zumab. In 2019, PRL3-zumab moved into a Phase II drug efficacy clinical trial at National Cancer Center Singapore (NCCS)[22]. Recently, PRL3-zumab has been promptly approved by the U.S. Food and Drug Administration (FDA) and National Medical Product Administration (NMPA) for a Phase II IND trial to treat all PRL3 solid cancers in the USA[23] and in China.

Apart from using PRL3-zumab to treat PRL3-positive tumors, in this report, we proposed PRL3-zumab could serve in another critical usage as an 'Adjuvant immunotherapy' immediately after patient's tumor removal to clean up PRL3+ PGCC stem-like cells. Therefore, we anticipate PRL3-zumab could act as 'Double Swords': not only inhibiting tumor growth but also preventing cancer relapse to overcome cancer metastasis.

## Results

### PRL3 overexpression induces the spontaneous formation of stem-cell like PGCCs. The Chinese Hamster Ovary (CHO) cell

line has been reported to possess a relatively stable karyotype which can form heteroploid cells upon virally or chemically induced fusion[24,25]. Using the CHO cell line, we stably over-expressed either myc-tagged PRL3 (CHO-myc-PRL3) or GFP-tagged PRL3 (CHO-PRL3). Both PRL3 overexpressing cell lines consistently displayed stark accumulation of giant cells approximately 5–10 times larger than surrounding mononucleated cells (Fig. 1a, b, Supplementary Fig. 1a). These giant cells, which are often multinucleated (Fig. 1a, white arrows), are reminiscent of polyploid giant cancer cells (PGCCs in green), a cancer stem cell-like population in human cancers[26]. Percentage of polypoid giant cells is 5 times higher in CHO-PRL3 cells compared to control (Supplementary Fig. 1b). To validate whether PRL3-induced PGCCs also possess stem cell characteristics, we investigated the expression of SOX2 and OCT4, two well-established embryonic stem cell markers[27,28]. PRL3-induced PGCCs had higher expression of both SOX2 in red (Fig. 1c) and OCT4 in red (Fig. 1d) than mononucleated cells. Quantification of fluorescent intensity indicated PRL3, SOX2, and OCT4 expression are highest in PGCCs compared to mononucleated cells (Supplementary Fig. 2). Therefore, PGCCs induced by PRL3, like other previously described PGCCs[18,26], also possess stem cell-like attributes.

### PRL3 promotes PGCC formation by causing incomplete cytokinesis despite the accumulation of DNA double-strand breaks (DSBs). PGCC formation can occur via endoreduplication (incomplete mitosis), endomitosis, or cytokinesis failure[29]. These phenomena result in endoreplication, whereby genetic material is copied in the absence of cell division, resulting in the accumulation of multiple or enlarged nuclei within cells.

To examine how PRL3 induces PGCC formation, we used real-time confocal imaging to follow the division of CHO-PRL3 cells. We observed various incomplete cytokinesis events in both mononucleated and multi-nucleated CHO-PRL3 cells grown under standard culture conditions. These predominantly exhibited cytokinesis failure as PRL3-overexpressing cells appeared to complete mitosis but failed to separate completely and formed polynucleated cells (Fig. 2a, Supplementary Movie 1).

Indeed, no significant difference in cell numbers was observed between control and PRL3-overexpressing cells grown over 48 h, $P = 0.09$ (Fig. 2b), implying that PRL3 overexpression could induce the spontaneous formation of PGCCs via uncoupling of cell cycling from DNA replication to promote polyploidy, resulting in daughter cells with massively increased ploidy rather than increased cell counts. As cell cycling appeared highly active in these cells, we examined the expression of several well-established cell cycle regulators.

PRL3-overexpressing cells were found to have elevated levels of various cell cycle activators including CDK2, CDK1, cyclin D1, and phosphorylated Aurora B compared to control cells (Fig. 2c). In contrast, PRL3-overexpressing cells demonstrated a clear downregulation of the cell cycle inhibitor, p15INK4B (Fig. 2c). These changes correlated well with the hyper-activated cell cycle profile described for PGCCs which, despite cycling less frequently[13], have been reported to possess constitutive cell cycle activation[26,30].

Genome instability can promote polyploidy[31], implicating incomplete cytokinesis as a possible adaptation to genotoxic stress. DNA double-stranded breaks (DSBs), if left unrepaired, can lead to chromosomal aberrations, genomic instability, and cell death[32]. The formation of DSBs triggers activation of many downstream effectors, including phosphorylation of the histone variant H2AX to form γH2AX (140S), which formed when DSBs appear. To investigate the genomic integrity of PRL3-

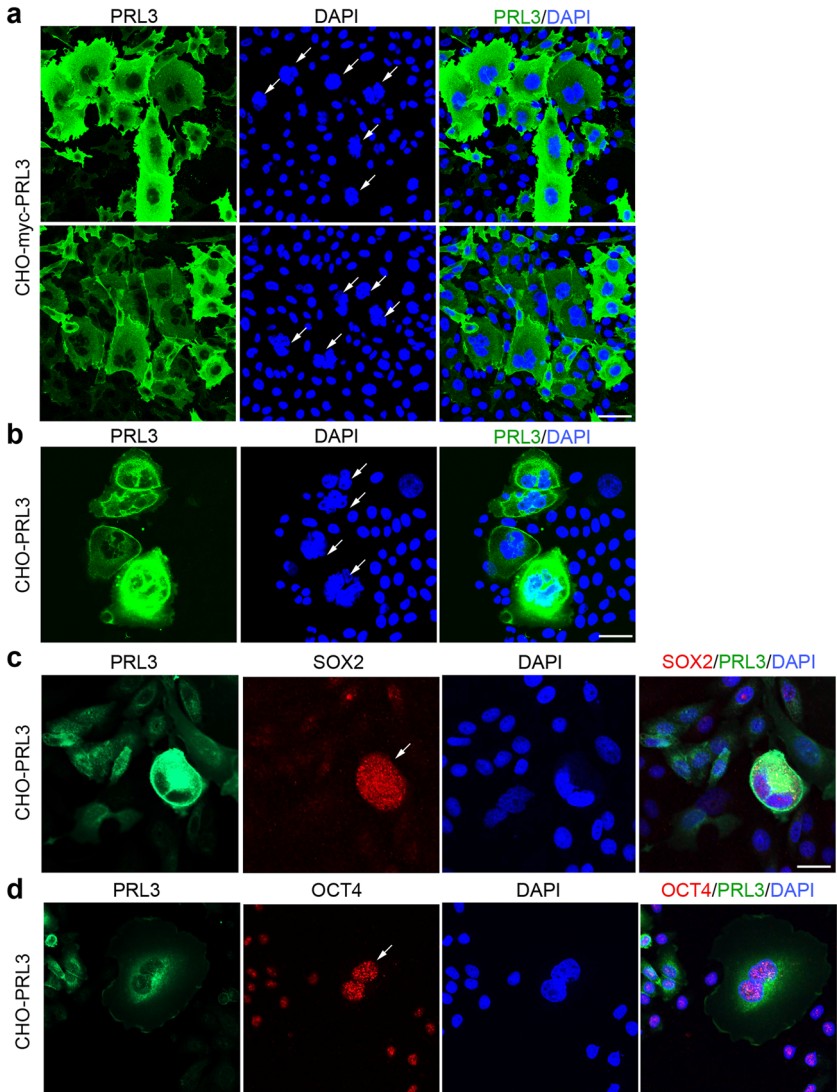

**Fig. 1 PRL3 overexpression induces the formation of stem cell-like polyploid giant cells. a**, **b** Immunofluorescence analysis of CHO cell lines stably expressing (**a**) myc-tagged PRL3 or (**b**) GFP-tagged PRL3 (CHO-PRL3). White arrows indicate multinucleated cells overlapped with high PRL3 expression. Cells in **a** were visualized via indirect staining with anti-PRL3 antibody (clone 318). Bar, 50 μm. **c**, **d** Immunofluorescence analysis of CHO-PRL3 cell pools with antibodies against (**c**) SOX2 and (**d**) OCT4. White arrows indicate multinucleated giant cells with elevated expression of both embryonic stem cell markers compared to mononucleated cells. Bar, 50 μm.

overexpressing cells, we compared the expression of γH2AX in CHO-PRL3 and control CHO cells. The nuclei of PRL3-overexpressing cells had clearly elevated γH2AX staining compared to control cells, even in normal culture conditions (Fig. 2d), and the accumulation of DSBs was greatest in PRL3-induced PGCCs (Fig. 2e), suggestive of extensive genome damage in these cells. Quantification of fluorescent intensity indicated γH2AX expression was highest in CHO-PRL3-PGCCs compared to mononucleated cells (Supplementary Fig. 3).

Unrepaired DSBs induce genome instability and promote tumorigenesis. One of the most important kinases activating cell cycle checkpoints following DNA damage is ataxia telangiectasia mutated (ATM). ATM is a major physiological mediator of H2AX phosphorylation in response to DSB formation and phosphorylates many other substrates including the checkpoint proteins, Chk1 and Chk2[33,34]. Chk2 is phosphorylated on the priming site T68 by ATM[35]. Since PGCCs have been reported to demonstrate chemo-resistance in human cancers and can give rise to therapy-resistant progeny with enhanced tumorigenicity[26,36], we hypothesized that

recurrent tumors might be enriched with cells harboring extensive genome damage derived from parental PGCCs. To test this hypothesis, we analyzed the expression of PRL3, γH2AX, and Chk2 in four pairs of post-chemotherapy primary tumors and patient-matched metastatic or recurrent tumors; Pair 1: *breast* invasive carcinoma sample, and its metastatic tumor in the *brain*; pair 2: *colon* adenocarcinoma sample and its metastatic tumor in the *liver*; pair 3: *rectal* cancer sample and its metastatic tumor in the *lung* and pair 4: *thyroid* cancer sample and its recurrent *thyroid* tumor. PRL3 expression was detected in all metastatic/recurrent tumors and primary tumors of pairs 3 and 4. Importantly, markedly elevated expression of γH2AX was seen in metastatic/recurrent tumor samples of pairs 1–4, but not detected in primary tumors apart from the primary tumor of pair 4. Elevation of phosphory-lated ChK2 (pChk2) was also detected in metastatic tumor of pairs 1–3 but not detected in recurrent tumor of pair 4. Both γH2AX and pCHK2 were upregulated in metastatic tumor of pairs 1–3 but not detected in the primary tumor. Additionally, the expressions were also correlated with PRL3 expression (Fig. 2f).

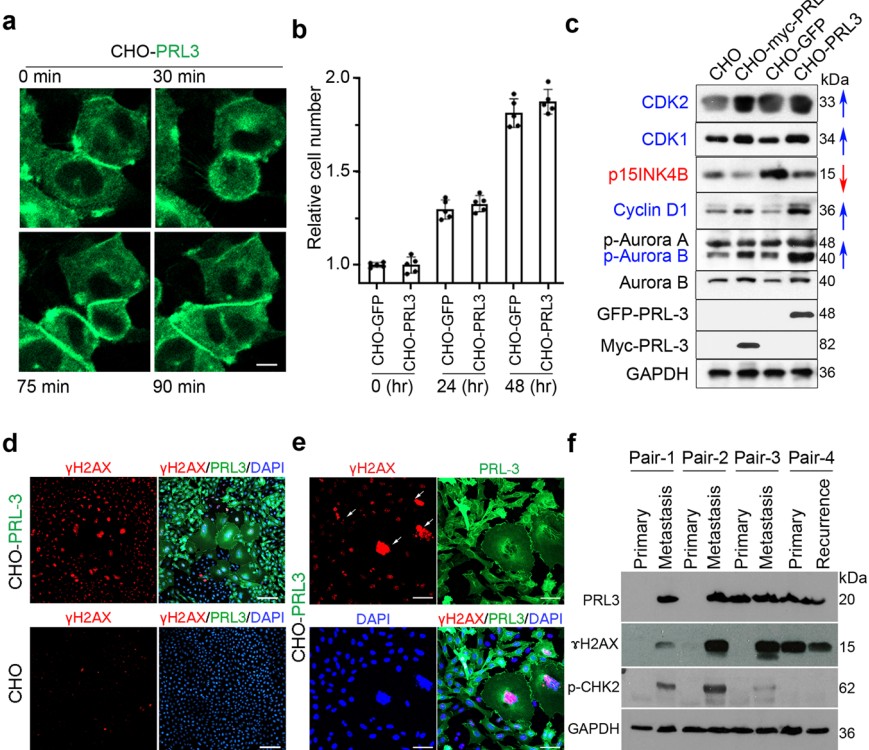

**Fig. 2 PRL3 enhances incomplete cytokinesis events despite the accumulation of DNA double-strand breaks. a** Representative real-time images of incomplete cytokinesis in CHO-PRL3 cells resulting in multinucleated PGCC formation. Bar, 50 μm. **b** No significant difference between the relative proliferation rate of CHO-GFP versus CHO-PRL3 cells over 48 h as measured using an MTS assay. Results represented as mean ± SD. Data obtained from five biological replicates. $P = 0.09$, Two-way ANOVA. **c** Unsynchronized CHO stable cell pools were assayed for various cell cycle-related proteins. Blue arrows indicate PRL3-induced cell cycle activators, whereas red arrows indicate PRL3-suppressed cell cycle inhibitors. GAPDH served as a loading control. **d**, **e** DNA double-stranded breaks (DSBs) visualized using immunofluorescence of CHO-PRL3 and CHO cells. Bar, 100 μm (**d**) or CHO-PRL3 cells. Bar, 40 μm (**e**) with anti-phospho-H2AX (γH2AX) antibody. White arrows indicate enlarged nucleus with elevated expression of γH2AX. Note the extensive accumulation of DSBs in the CHO-PRL3 cell pool compared to CHO cells. **f** Western blot of four pairs of human primary tumors after chemotherapy and matched distant metastasis or recurrent tumor from the same patient. PRL3 overexpression correlates with intense phosphorylation of ɣH2AX in all four pairs and with p-Chk2 in three pairs of metastatic cancer.

Pairs 1–3 are primary and metastatic tumors whereas pair 4 is primary and recurrent tumor.

The nature of recurrent tumor might be the same as primary tumor specifically for this pair 4 only and does not follow our hypothesis, expressing γH2AX in primary tumor and not expressing pChK2 in recurrent tumor.

These results support our in vitro observations and suggest that metastatic/recurrent tumors, which have upregulated PRL3 expression are able to survive and resist cell death despite extensive DNA damage.

**PRL3 induces chemo-resistant PGCC formation by suppressing the ATM DNA damage-signaling pathway.** Since PGCCs are highly resistant to chemotherapeutic drugs[18], we investigated the possibility that PRL3-induced PGCCs could avert cell cycle arrest and apoptosis upon genotoxic stress induced by chemotherapy. Whereas treatment of parental CHO cells with cisplatin resulted in widespread cell detachment and death within 7 days, CHO-PRL3 cells remained firmly attached to the culture vessels at day 11 (Fig. 3a). Intriguingly, some of these cells grew up to multiple times larger than untreated mononucleated cells (Supplementary Fig. 4). Quantitation of mean nuclear areas at day 11 revealed a 4-fold increase in CHO-PRL3 cells ($414 \pm 292$ μm²) over scantily surviving CHO cells ($113 \pm 51$ μm²), $P < 0.0001$ ($t$ test), the latter being similar to untreated cells at day 0 (Fig. 3b).

For cancer cells to bypass mitotic catastrophe and enter into polyploidy with unrepaired cisplatin-induced DNA double-stranded breaks requires uncoupling of the spindle checkpoint from apoptosis[37]. Since ATM kinase phosphorylates various proteins involved in cell cycle checkpoint control, apoptosis, and DNA repair, we anticipated that PRL3-overexpressing cells might generally downregulate pro-apoptotic ATM signaling in cisplatin-treated cells to permit PGCC survival. Compared to control cells, levels of phosphorylated ATM (Ser1981) were suppressed in CHO-PRL3 cells after 4 and 11 days of cisplatin treatment (Fig. 3c, lanes 8 and 12). Suppression of ATM activity was mirrored by a reduction in phosphorylation of the ATM substrate, p53, on Ser15 (Fig. 3c, lanes 8 and 12). Since ATM inactivation appeared to be a dysregulated target of PRL3 to endow resistance and promote PGCC accumulation upon cisplatin-induced DNA damage, we compared PGCC formation in control CHO and CHO-PRL3 cells exposed to prolonged periods of cisplatin. Whereas CHO-PRL3 cells accumulated significantly more PGCCs than control cells upon cisplatin treatment alone, $P = 0.0002$ (Fig. 3d), These results demonstrate that suppression of ATM-mediated DNA damage signaling pathway might be important for PGCC survival under prolonged chemotherapeutic stress. Herein, we propose a model wherein prolonged cisplatin selection of PRL3-expressing cells results in the accumulation of surviving PGCCs with robust chemo-resistance (Fig. 3e).

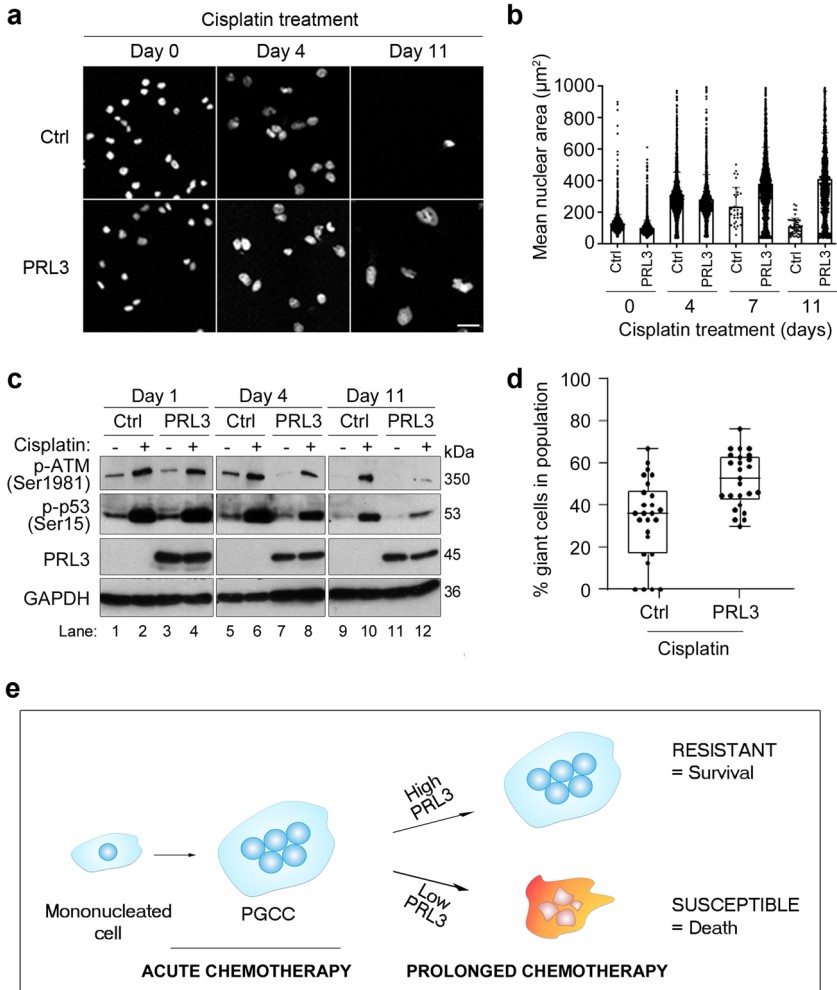

**Fig. 3 PRL3 promotes chemo-resistance by inducing PGCCs formation. a** Phase-contrast images demonstrating enhanced CHO-PRL3 ('PRL3') cell survival after 11 days of cisplatin treatment compared to CHO-GFP ('Ctrl') cells. Bar, 100 μm. **b** Graph showing changes in the mean nuclear area of cells (**a**) over the course of cisplatin treatment. Results presented as mean ± SD. $P = 0.05$ (two-way ANOVA), $P < 0.0001$ PRL3 vs. Ctrl at Day 11 (Student's $t$-test) (**c**) Western blot analysis of cells left untreated or treated with 5 μg/mL cisplatin for up to 11 days. GAPDH served as a loading control. **d** Chart presenting the mean percentage (%) of giant cells in the population of cisplatin-treated Ctrl and PRL3-overexpressing cells. $n = 26$ per group; $P = 0.0002$, Student's $t$-test. **e** A hypothetical model showed how PRL3 promotes resistance during prolonged cisplatin treatment by enhancing PGCC survival.

**PRL3-zumab targets PRL3$^+$ PGCCs and reduces tumor relapse**. PGCCs have also been shown to endow resistance to several classes of targeted therapies including the proteasome inhibitor, bortezomib[38], as well as the tyrosine kinase inhibitor, BMS-777607[39]. Since PGCCs constitute a key subpopulation of stem-like cancer cells that are well-positioned to the rapid evolution of patient tumors during conventional chemotherapy and radiotherapy regimes, as well as targeted therapies, often leading to deadly disease recurrence in cancer patients, it is of profound importance to identify modalities that can target malignant subpopulation to reduce cancer morbidity.

Immunotherapy is revolutionizing the clinical management of multiple tumors. Recently, we demonstrated that immunotherapy with PRL3-zumab, a humanized PRL3 antibody, could successfully inhibit tumors expressing the PRL3 oncoprotein in mouse models to prevent cancer metastasis[4,21]. In this study, we developed a much more aggressive 'tumor removal & relapse' mouse model using the rapidly growing naturally occurring PRL3-positive B16F0 mouse melanoma cell line (Supplementary Fig. 5, Lane 1) which has 2.6 ± 0.8% of PGCCs in normal culture conditions (Supplementary Fig. 6). We inoculated the B16F0 cells into both flanks of mice, waited 2 weeks for primary tumors to

develop (Fig. 4a–i and 4b–i, upper panel), then removed the tumors (Fig. 4a–i and 4b–i, lower panel). The mice were then divided into two groups: the 'Untreated' group with placebo and the 'Treated' group with PRL3-zumab. After 2 weeks, relapsed tumor sizes were monitored and compared between these 2 groups. In the 'Untreated' group, the mean tumor volume was 0.6 ± 0.11 cm$^3$ in primary tumor and 1.25 ± 0.22 cm$^3$ in relapsed tumors, more than 2 times bigger than primary tumors, $P = 0.053$ (Fig. 4a–ii, iii). In contrast, the mean tumor volume in the 'Treated' group was 0.51 ± 0.8 cm$^3$ in primary tumor and 0.02 ± 0.01 cm$^3$ in relapse tumors, $P = 0.003$, more than 20 times smaller than primary tumors (Fig. 4b–ii, iii).

The above 'tumor removal & relapse' model with B16F0 (PRL3 positive) mouse melanoma cancer cell line was further compared with our other routine standard model using a naturally occurring PRL3 positive SNU-484 human gastric cancer cell line (Supplementary Fig. 5, Lane 3) which has 1.27 ± 1.03% of PGCCs in normal culture condition (Supplementary Fig. 6). Similarly, SNU-484 cancer cells were implanted in both flanks of the mice, after waiting 2 weeks for the primary tumor to develop (Fig. 4c–i and d–i, upper panel), the primary tumors were surgically removed (Fig. 4c–i and d–i, lower panel). The mice were then

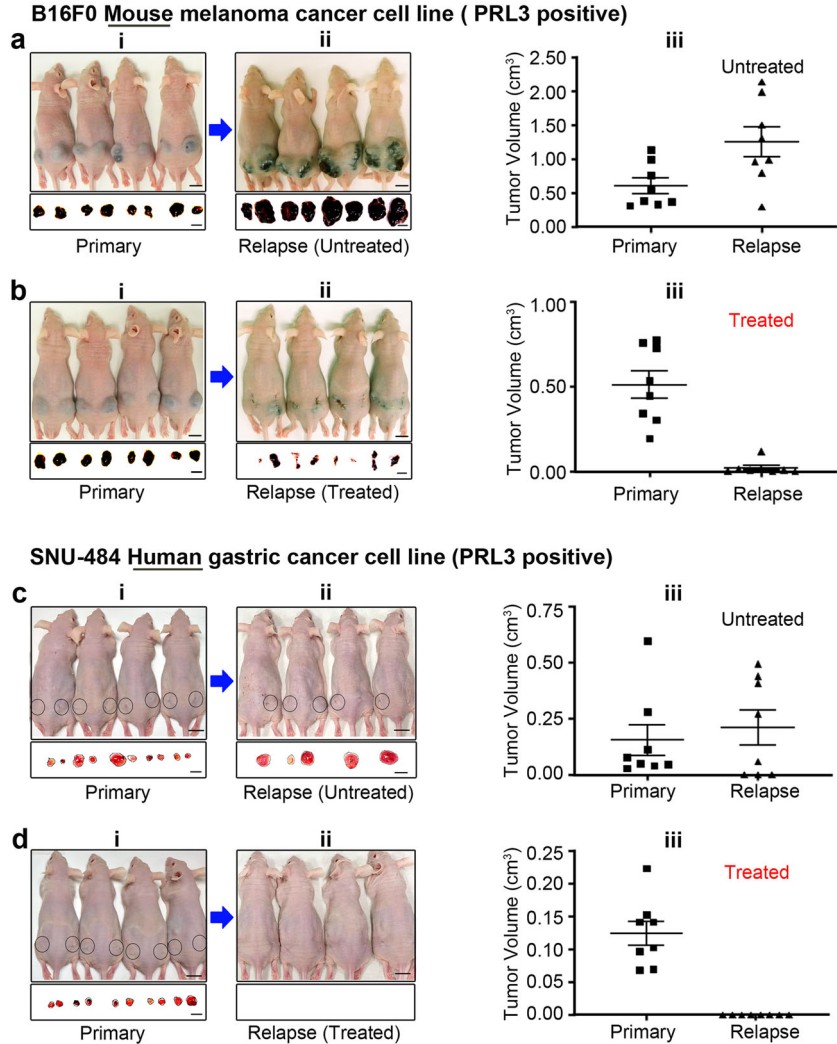

**Fig. 4 PRL3-zumab, as an 'Adjuvant Immunotherapy' after tumor removal, prevent tumor relapse. a, b** Mice with xenograft tumors formed by B16F0 (PRL3 positive) cells: 2 weeks after inoculation (**a-i**, **b-i** upper panels) and removed primary tumors in black (**a-i**, **b-i**, lower panels). Mice with relapse tumors formed 2 weeks after tumor removal in the untreated group (**a-ii**) and treated group (**b-ii**). Tumor volume of B16F0 primary (**a-i**) and relapse tumors (**a-ii**) from untreated group, $n = 8$, data represents mean ± SEM. $P = 0.053$ Student's $t$-test (**a-iii**). Tumor volume of B16F0 primary (**b-i**) and fewer and smaller relapse tumors (**b-ii**) from treated group, $n = 8$, data represent mean ± SEM. $P = 0.003$ Student's $t$-test (**b-iii**). **c, d** Mice with xenograft tumors formed by SNU-484 (PRL3 positive) cells: 2 weeks after inoculation (**c-i**, **d-i** mice image at upper panel) and removed primary tumors (**c-i**, **d-i**, tumor images at lower panel). 2 weeks after tumor removal from untreated group (**c-ii**, mice image at the upper panel and removed tumor at the lower panel), and PRL3-zumab treated group (**d-ii**, mice image at the upper panel and no tumor relapse at the lower panel. Tumor volume of SNU-484 primary (**c-i**) and relapse tumors (**c-ii**) from the untreated group, $n = 8$, data represent mean ± SEM. $P = 0.45$ Student's $t$-test (**c-iii**). Tumor volume of SNU-484 primary (**d-i**) and no relapse tumors (**d-ii**) from treated group, $n = 8$, data represents mean ± SEM. $P < 0.0002$ Student's $t$-test (**d-iii**). Black circle over the mouse indicates tumor location. Bar, 10 mm.

divided into 'Untreated' (Placebo) and 'treated' (PRL3-zumab) groups. The mean primary tumor size of the untreated group (Fig. 4c–i, bottom panel) was $0.15 ± 0.07$ cm³ whereas the mean relapsed tumor size (Fig. 4c–ii, bottom panel) was $0.21 ± 0.08$ cm³, much bigger than the primary tumors, $P = 0.45$ (Fig. 4c–iii). In contrast, in the 'Treated' group, the primary tumor size (Fig. 4d–i) was $0.12 ± 0.2$ cm³ and there was no tumor relapse, $P < 0.0001$ (Fig. 4d–ii, iii). Both B16F0 and SNU-484 'tumor removal and relapse' experiments were repeated (Supplementary Figs. 7 and 8) and confirmed reproducible results.

To investigate if PRL3-zumab prevents tumor repopulation after surgery in these mice by eradicating rapidly evolving PGCC-like cells, we analyzed the difference in cell morphology and DNA content in frozen tumor sections derived from both the original ("primary") and post-surgical relapsed tumors ("relapse") of the animal models we used in Fig. 4, B16F0 and SNU-484 tumors. In the B16F0 tumor model, compared to primary tumors (Fig. 5a–i), cancerous cells in relapsed tumors appeared to be enriched with PGCCs as inferred by the high frequency of cells with enlarged nuclei stained with DAPI (Fig. 5a–ii). Quantitation of nuclear areas confirmed a significantly larger mean nuclear size in relapse tumor cells compared to primary tumor cells, $P = 0.0119$ (Fig. 5a–iii). Similar results were obtained in the SNU-484 tumor model as enlarged hyperchromatic nuclei can be seen in relapse tumors compared to control (Fig. 5b–i, ii). Quantitation of nuclear area clearly confirmed that the nuclei of relapsed tumor cells were larger than primary tumor cells, $P = 0.0086$ (Fig. 5b–iii).

These findings implied that PGCC-like cells with larger nuclei (or increased DNA content) were preferentially enriched in relapsed tumors. The infrequent cell cycle is a well-established

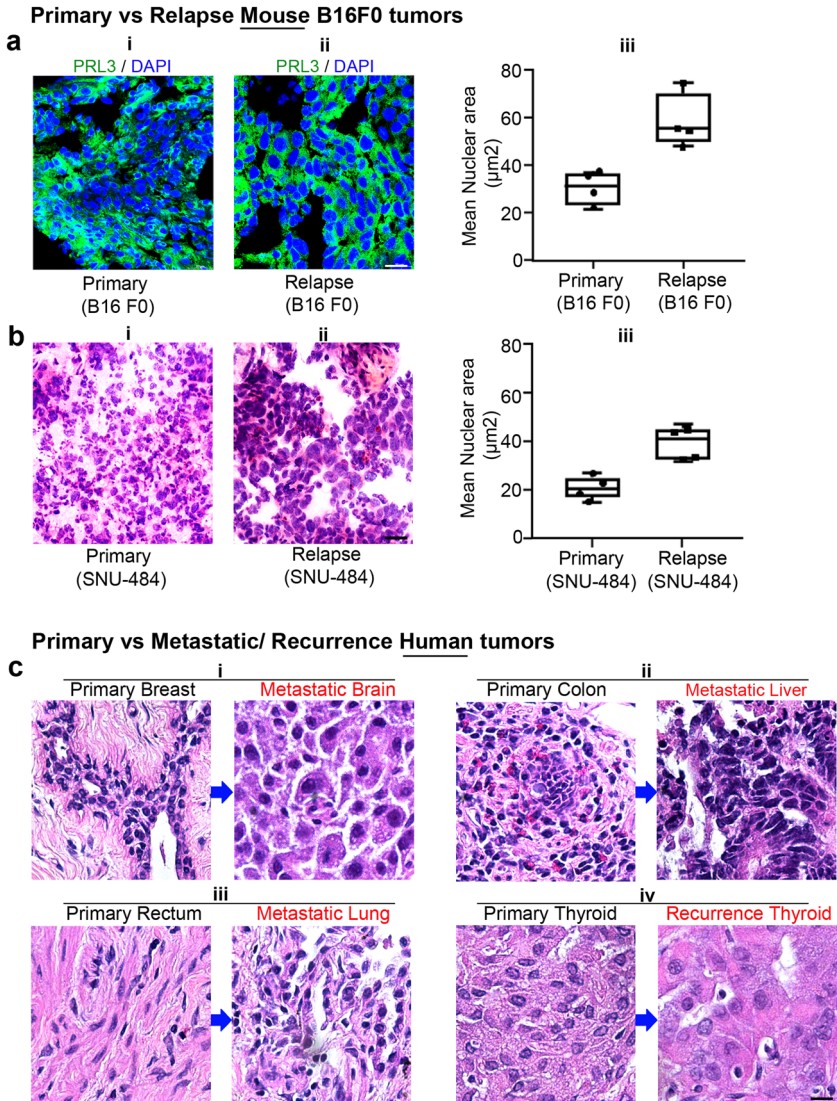

**Fig. 5 PRL3⁺ PGCCs were enriched in relapsed tumors. a** Representative immunofluorescence image of tumor sections from B16F0 primary (**a-i**) and relapsed tumors (**a-ii**). PRL3 protein is shown in green in the cytoplasm and DAPI staining nucleus in blue. Note enlarged nuclei were stained with DAPI in relapsed tumors compared to primary tumors. Bar, 20 μm. Box-and-whisker plot of mean nuclear areas of cells from B16F0 primary and relapse tumors (**a-iii**). Data were collected from a mean nuclear area of four sets of primary and relapse B16F0 tumors. $P = 0.0119$, Student's $t$-test (**a-iii**). **b** Representative H&E-stained images of tumor sections from SNU-484 primary (**b-i**) and relapsed tumors (**b-ii**). Note the hyperchromatic, enlarged nuclei in relapsed tumors compared to primary tumors. Bar, 20 μm. Box-and-whisker plot of mean nuclear areas of cells from SNU-484 primary and relapse tumors (**b-iii**). Data were collected from a mean nuclear area of four sets of primary and relapse SNU-484 tumors. $P = 0.0086$, Student's $t$-test (**b-iii**). **c** H&E-stained sections of four pairs of the human primary tumor after chemotherapy and metastatic or recurrent tumors from same patients. PGCCs, hallmarked by abnormal accumulation of nuclear content, can be detected as hyperchromatic enlarged nuclei in metastatic or recurrent tumor sections (**c-i, ii, iii, iv**). Bar, 50 μm.

resistance mechanism against cytotoxic insult. PGCCs are often generated upon exposure to chemotherapeutics[40–42], inducing the accumulation of tumor cells with elevated ploidy. By increasing chromosome numbers, PGCCs provide a mechanism to generate infrequently cycling tumor cells, establishing a general resistance mechanism against cytotoxic chemotherapy treatments designed to target otherwise actively cycling tumor cells[43].

To translate our animal findings to clinical relevance, we performed hematoxylin and eosin (H&E) staining in tumor sections from the four patients matched fresh-frozen tumor pairs described earlier in Fig. 2f, who had metastasis/recurrent tumor that appeared 6–15 months after chemotherapy to treat the primary tumors (Table 1). Patient 1 (pair 1)—primary breast cancer to brain metastasis; patient 2 (pair 2)—primary colon adenocarcinoma to liver metastasis, patient 3 (pair 3)—primary rectal cancer to lung metastasis, and patient 4 (pair 4)—primary thyroid carcinoma and thyroid carcinoma recurrence) (Fig. 5c–i–iv). In line with our findings from our animal tumor 'removal and relapse model', Fig. 5a and b, we observed PGCC accumulation in metastatic/recurrent tumor sections (Fig. 5c in red headlines). These findings from precious human samples are consistent with our basic research findings in vivo and in vitro, indicating that PGCCs are increased in chemo-resistant metastasis/relapse tumors.

## Discussion
This study herein reports a clinically relevant tumorigenic role of PRL3 oncoprotein vis-à-vis the induction of PGCC formation.

**Table 1 Clinicopathological details of fresh-frozen human samples analyzed in this study.**

| Human tissue | | | Duration of chemotherapy prior to sample collection | Used in |
|---|---|---|---|---|
| Patient 1 (Pair-1) | Primary | Breast invasive carcinoma | 6 months | WB, H&E |
| | Metastatic | Breast invasive carcinoma distant metastasized to the brain | – | WB, H&E |
| Patient 2 (Pair-2) | Primary | Colon adenocarcinoma | 15 months | WB, H&E |
| | Metastatic | Colon adenocarcinoma distant metastasized to liver | – | WB, H&E |
| Patient 3 (Pair-3) | Primary | Rectal adenocarcinoma | – | WB, H&E |
| | Metastatic | Rectal adenocarcinoma adenocarcinoma distant metastasized to lung | – | WB, H&E |
| Patient 4 (Pair-4) | Primary | Thyroid carcinoma | – | WB, H&E |
| | Recurrence | Thyroid carcinoma | – | WB, H&E |

WB western blotting, H&E hematoxylin-eosin staining.

PRL3-overexpression was sufficient to increase the frequency of incomplete cytokinesis events driving polyploidy and formation of hypertrophic PGCCs, often with multiple nuclei. PRL3-induced PGCCs expressed the embryonic stem cell markers SOX2 and OCT4 and were resistant to genotoxic stress induced by cisplatin treatment. However, PRL3+ PGCCs could be targeted successfully using PRL3-zumab immunotherapy to reduce tumor relapse in our mouse model. As recurrent human cancers are enriched with PGCCs, our findings provide insights into the mechanism for PRL3 in promoting drug resistance and tumor relapse. Our findings open up the possibility of two distinct therapeutic avenues for our First-in-Class PRL3-zumab antibody drug: 1. potentially targeting clinically resistant tumor sub-population. 2. Serving as an 'adjuvant immunotherapy' after tumor removal surgery to clean up PRL3+ cells from the circulation and thereby prevent tumor relapse and metastasis. The critical ability of PRL3-zumab to eradicate PGCC is the key against cancer metastasis.

Most chemotherapeutic drugs aim at disrupting the mitotic process to kill rapidly dividing cancer cells. However, they also cause damage to normal tissues and result in renal toxicity, neurotoxicity, myelosuppression, and peripheral neuropathy[44]. To balance tumor-specific toxicity and unwanted side effects, there are often long waiting periods between chemotherapeutic rounds, providing an opportunity for the repopulation of cancer cells between treatment intervals[45]. PGCCs, which cycle less frequently, are especially adept at entering a reversible senescent state, called therapy-induced senescence (TIS), to allow cancer cells to avoid the genotoxic effects of chemotherapy and radiotherapy[46]. Polyploidy is independently predictive of poor relapse-free survival[47], suggesting that polyploidy may promote tumorigenesis leading to tumor relapse. In order to investigate the relationship between PRL3 expression and polyploidy, we examined matched pairs of post-chemotherapy primary tumors and their metastatic relapse tumors and found that metastatic relapse samples were enriched in PGCCs, indicating that PRL3 may have a role in promoting polyploidization in such tumors. In PRL3-overexpressing cells, we also found that suppression of the ATM-mediated DNA damage-signaling pathway endowed cells with DNA damage insensitivity and the ability to escape from apoptosis despite prolonged genotoxic stress induced by cisplatin treatment, leading to the accumulation of γH2AX levels. Interestingly, increased γH2AX level measurements have been suggested to help pre-cancerous lesions or cancer at its early stages[48], supporting the early role of PGCCs in promoting tumorigenesis vis-à-vis DNA replication stress leading to DNA DSBs, genomic instability, and selective pressure for tumor-driving mutations[49]. Importantly, since mitosis-independent PGCCs possess the ability to exit TIS via de-polyploidization to generate the seed population for tumor repopulation and disease recurrence[2,9], our finding that PRL3-zumab can inhibit the recurrence of PRL3+ tumor which is possibly hyperploid in nature and potentially represent a step forward in targeting tumor 'dormancy' by abolishing this classically chemoresistant, residual self-renewing PGCC population.

Previously, we and others have demonstrated the value of PRL3 as an excellent oncotarget[4,21,50–54]. We further showed that PRL3 was expressed in ~85% of fresh-frozen gastric tumor tissues, but not in patient-matched normal gastric tissues[21]. Since elevated PRL3 expression has been described in many other tumors types[50], we sought to broadly characterize PRL3 protein expression in 151 hard-to-obtain, fresh-frozen patient tumor samples from 11 different cancer types, including several aggressive malignancies in our previous study. Importantly, PRL3 is frequently (80.6%) overexpressed in multiple human cancers[4]. PRL3-zumab could potentially serve as a broad cancer drug against multiple cancer types. In 2018, a Phase I clinical study of PRL3-zumab was successfully completed in Singapore[55]. Currently, PRL3-zumab is undergoing Phase II clinical trials for its efficacy[22]. PRL3-zumab was recently approved by the FDA for Phase II IND clinical trial in the US for all solid cancer types with no required pre-diagnosis as PRL3 oncotarget has been highly (80.6%) overexpressed in multiple cancer types[23]. PRL3-zumab has been approved by NMPA to carry out IND Phase II clinical trial in China. Taken together, our studies suggest that PRL3 plays a role in tumor relapse by inducing polyploidy and PGCCs formation, leading to tumor relapse. This is therefore a timely utility for PRL3-zumab in blocking PGCCs-driven tumor recurrence by targeted inhibition of PRL3 for the urgent use in inhibiting tumor growth and more importantly, to reduce cancer metastasis and tumor relapse.

## Methods

**Cell lines and treatments.** CHO-K1 cells (CCL61) and B16F0 cells (CRL6322) were obtained from ATCC. SNU-484 cells (CVCL_0100) were obtained from Korean Cell Line Bank (KCLB). CHO cell lines stably expressing myc-PRL3, EGFP-PRL3 (CHO-PRL3), and EGFP-PRL3-C104S (CHO-GFP) have been described previously[56]. The experiments were performed between 3 and 5 passages of cell lines. Periodic mycoplasma testing was performed using an EZ-PCR mycoplasma test kit (Biological Industries). Cells were maintained in RPMI media supplemented with 10% fetal bovine serum (FBS) at 37 °C under 5% carbon dioxide ($CO_2$) atmosphere. Cell proliferation assay of CHO-GFP and CHP-PRL3 cells was done by using the MTS assay kit (Promega). For cisplatin treatment, a preliminary study was done using 1, 5, and 10 μg/mL of cisplatin and choose 5 μg/mL as it can give the optimal result. Cells were treated with cisplatin (5 μg/mL; Hospira) for the indicated durations in complete growth media. The cell culture media was changed every 48 h together with the drug.

**Antibodies.** SOX2 (D6D9), OCT4 (C30A3), γH2AX (Ser139), CDK1, CDK2, p15INK4B, Cyclin D1 (DSS6), p-Aurora A (T288), p-Aurora B (T232, 1:1000), Aurora B, c-Myc (9E10) and Phospho-Chk2 (Thr68), Phospho-ATM (Ser198), Phospho p53 (Ser 15) antibodies were purchased from Cell Signalling Technology.

GFP (clone B-2) antibody was purchased from Santa Cruz Biotechnology. Primary antibodies are used in 1:200 dilution for Immunofluorescence assay and 1:1000 dilution for Western blotting. GAPDH (clone MAB374, 1:4000) antibody was purchased from Millipore.

Horse radish peroxidase (HRP)-conjugated goat anti-mouse and anti-rabbit secondary antibodies (1:5000) were purchased from Jackson ImmunoResearch. Alexa Fluor 488 goat anti-mouse antibody and Alexa Fluor 568 goat anti-rabbit antibody (1:200) were purchased from Life Technologies. Anti-PRL3 monoclonal antibody (mAb) (clone 318, 1:200 for Immunofluorescence and 1:2000 for western blot) was generated in-house. PRL3-zumab was engineered based on the original framework of murine anti-PRL3 mAb (clone 318).

**Western blot analysis.** Protein extraction and immunoblotting were performed[21]. For tissues, the excised samples (5 mm³) were suspended in RIPA lysis buffer supplemented with a protease and phosphatase inhibitor cocktail (Roche) for 15 min at 4 °C and disrupted completely with a tissue homogenizer (Polytron). Lysates were clarified by centrifugation at $13,000 \times g$ for 40 min at 4 °C. For cultured cells, $5 \times 10^6$ cells were lysed in lysis buffer and clarified as above procedure. Protein concentrations of both tissue and cell lysates were estimated using a bicinchoninic assay kit (Pierce). Samples were boiled after adding 2× Laemmli buffer containing DTT (50 mM final concentration) and used immediately for Western blotting or stored at −80 °C until use.

Tissue (40 µg) or cell lysates (200 µg) were resolved in 12% SDS–polyacrylamide gels, transferred to nitrocellulose membranes followed by blocking and probing with the indicated primary antibodies overnight at 4 °C. The membrane was incubated with the respective HRP-conjugated secondary antibodies for 1 h after thorough washing with tris buffered saline-tween (TBS-T) buffer (20 mM Tris pH 7.6, 140 mM NaCl, 0.2% Tween-20). After washing again with TBS-T buffer, and the membrane was visualized using a chemiluminescent substrate (Pierce).

**Immunofluorescence.** Cells grown on coverslips were fixed in 4% paraformaldehyde followed by simultaneous permeabilization and blocking with 5% goat serum (Sigma) supplemented with 0.3% Triton X-100 in 1× phosphate-buffered saline (PBS). The coverslips were then incubated with antibodies (SOX2 or OCT4, γH2AX) overnight at 4 °C, washed, and incubated at room temperature for 1 h with appropriate secondary antibodies. For tumor sections, B16F0 tumors were sectioned into 10 µm slices using a cryostat (Leica) at 16 °C. Fixation, permeabilization, and blocking procedures were performed the same as above. Tissue sections were incubated with PRL3 mouse antibody (m318, in house) overnight at 4 °C, washed, and incubated at room temperature for 1 h with an appropriate secondary antibody. Coverslips and tissue section slides were then mounted in media containing DAPI and visualized using an LSM800 confocal microscope (Zeiss AG).

Quantification of fluorescence intensity was done using Image J software.

**Xenograft tumor recurrence model.** Animal study was approved by A*STAR Institutional Animal Care and Use Committee (IACUC) with the approval number (#IACUC 161130) and performed in accordance with approved guidelines. 8 weeks old, male Ncr/Nude mice (InVivos Pte Ltd., Singapore) were used in this study. $5 \times 10^5$ B16F0 cells or $5 \times 10^6$ SNU-484 cells in 150 µl of PBS were injected into both flanks of anesthetized mice. After 2 weeks, resultant tumors (>0.5 cm diameter) were surgically removed under anesthesia, and the mice were randomly divided into treated or untreated groups to receive biweekly doses of either PRL3-zumab (5 mg/kg) or placebo, respectively. Tumor recurrence was then monitored in both groups for 2 weeks after resection. Tumor volumes were calculated using the formula; $0.4 \times$ tumor length × tumor width².

**Scoring for PGCCs.** For cultured cells on coverslips, images of DAPI-stained nuclei were captured (>20 representative views per sample), measured, and automatically calculated using Zen software. a cell was defined as a PGCC if its nuclear area exceeded 350 µm² (or approximately three times the average mononuclear area) in size.

For tissue sections, images of DAPI-stained nuclei or H&E stained were captured (three representative views per sample), measured the nucleus area by image J software. The mean value was taken for each tumor and compared. $n = 8$ for primary and relapse B16F0 tumors & SNU-484 tumors.

**Real-time fluorescence imaging.** Images of GFP-expressing CHO-PRL3 cells in complete media were scanned at 15 min intervals over 3 days using an LSM800 confocal microscope equipped with an environmental chamber to maintain cells constantly at 37 °C under 5% CO₂ atmosphere. Movies were assembled from time-lapsed images using Zen software (Zeiss AG).

**Patient samples.** Details of the fresh frozen samples obtained from the National University of Singapore Tissue Repository (NUHS-TR) are provided in Table 1. Specimens were collected and stored in liquid nitrogen immediately after surgery. Written informed consent was obtained from all patients. The collection and use of human tissue samples were approved by the Institutional Review Board (IRB) of the National University of Singapore and the Institute of Molecular and Cell Biology, Singapore.

**Hematoxylin and Eosin (H&E) staining.** Was performed at the Advanced Molecular Pathology Lab, IMCB. Both human tumors and mouse tumors formed by human gastric cancer cells (SNU-484) were sectioned into 10 µm slices using a cryostat (Leica) at 16 °C, transferred onto poly-L-lysine-coated slides (VWR), fixed in 4% paraformaldehyde.

**Statistics and reproducibility.** Nuclear areas were assessed in randomly selected fields and analyzed with Student's $t$-test, one-way ANOVA, and two-way ANOVA. Significance was defined as $P < 0.05$, and all tests were two-sided. Statistical tests were performed with Prism 4.0 (GraphPad software). In the animal experiment, the group sizes were calculated based on power analysis using the following parameters: (1) differences in mean 50% (estimated according to previous experiments), (2) $P$ value 0.05, and (3) power 80%.

**Reporting summary.** Further information on experimental design is available in the Nature Research Reporting Summary linked to this paper.

## Data and materials availability

All data generated or analyzed during this study are included in this article and its supplementary information file. The original blot images for Fig. 1c, f and 3c are mentioned in Supplementary Fig. 9. Source data for Figs. 2–5, Supplementary Figs. 1–4 and 6–8 can be obtained in Supplementary Data 1. Raw image files will be made available by the corresponding author upon reasonable request.

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

## Acknowledgements

We are grateful to the National University Hospital Tissue Repository, Singapore, for providing clinical samples, as well as to the Advanced Molecular Pathology Laboratory (IMCB, A*STAR) for pathological analysis and H&E staining of clinical tumor samples. We thank Dr. Vivian Yujing Lim and Dr. Joel Xuan En SNG (IMCB, A*STAR) for their careful proof-reading of this manuscript. This work was supported by Open Fund-Individual Research Grant (OF-IRG), National Medical Research Council (NMRC) Singapore with the project ID of NMRC/OFIRG/0053/2017 and core fund provided by Institute of Molecular & Cell Biology, Agency for Science, Technology and Research (A*STAR), Singapore.

## Author contributions

M.T., A.Q.O.A., and Q.Z. designed the experiments and prepared the manuscript. M.T., Z.Y., J.Y.O., Q.X., A.Q.O.A., A.G., J.L., K.G., and K.H.A. performed the experiments and analyzed the results. M.T., A.Q.O.A., K.H.A., and Q.Z. proofread and finalized the manuscript. All authors approved the manuscript.

## Competing interests

The authors declare the following competing interests: Q.Z. is the founder of Intra-Immu SG Pte Ltd, a spin-off company granted licensing rights to the PRL3-zumab IP portfolio. The other authors declare no competing interests.
