## [Peer Review File · Communications Biology]

Reviewers' Comments:

Reviewer #1:

Remarks to the Author:

This manuscript has two main foci: the PRL3 and polyploid giant cancer cells (PGCCs) with a discussion of how expression of PRL3 may be related to PGCCs and how PRL3 may be related to recurrence. Utilizing constitutive overexpression in CHO cells, the authors show that PGCCs in the CHO cell line are PRL3 positive, and that those PGCCs express classical cancer stem-cell makers and are positive for gH2AX, a surrogate for double strand breaks. Next, evidence is presented that PGCCs are formed via failed cytokinesis. The link with chemotherapy resistance is made by presenting data that overexpressing PRL3 CHO cells (assumed to be PGCCs) have suppressed ATM signaling following long-term high-dose cisplatin treatment. In vivo data from two immunocompromised mouse models show that targeting PRL3 with PRL3-zumab reduces relapse following s.c. tumor removal; interestingly, data is presented that the "relapsed" tumors may be enriched for PGCCs. This data is consistent with prior published reports as well as presented H&E from patient tissue that recurrence and/or recurrence is enriched for PGCCs.

This paper is interesting and the data connecting PRL3 and PGCCs is new to the field, but the data presented do not support the conclusions or the title of the paper: specifically, there is not direct evidence that PRL3 induces PGCCs (necessarily a cancer cell phenomenon) nor that PRL3-zumab eliminates those PGCCs to prevent tumor relapse. Moreover, a number of necessary controls and necessary quantification are absent from the manuscript that limits interpretation of data. Separate from the lack of presented controls (presumably a simple revision to include), the data are sound. My recommendation is either to 1) revise the conclusions derived from these data to more accurately capture what data are presented or 2) include additional data to enable the reader (and this reviewer) to directly assess PRL3, PGCCs, and recurrence following chemotherapy. Specific points are listed below:

1. It is curious that in this study of polyploid giant cancer cells that the non-cancer cell line model CHO was chosen. While a valuable model, it would be valuable to see the impact of PRL3 overexpression in a cancer cell line model. This is especially important given the discordance between the overexpression PRL CHO cells showing a high level of PGCCs (e.g., Fig 1a) and the low level (and perhaps absence of) PGCCs in the PRL+ F16F0 and SNU-484 primary tumors (Fig 5a and bi). This is especially critical given the conclusion that PRL3 expression alone is sufficient to induce PGCC formation.
2. Appropriate negative controls for CHO-myc-PRL3 and CHO-PRL3 are absent throughout. Non-manipulated CHO cell line is not an appropriate control as PGCCs are well known to form upon transfection or lentiviral transduction in cell culture. A control such as the same promoter driving a marker would be more appropriate - this reviewer is unsure if that is the same as the "CHO-GFP" presented in Fig 2b and Fig2c. Images of such the control cell line should be included in Fig 1a, Fig 1b, Fig 1c, Fig 1d, Fig 2d, Fig 3a, Fig 3b, Fig 3c and Fig 3d.
3. A number of compelling images are presented throughout the manuscript, but quantification is lacking. For example, in Fig 1, PGCCs are shown via IF. What is the % PGCC of the population? What is the total number? Is it greater than control? Without this data, it is impossible to determine if PRL3 impacts PGCC formation or survival.
4. Similarly, it would be useful to include quantification of staining (e.g., PRL3, SOX2, OCT4, gH2AX), including normalization by cell area (for PRL3, SOX2, OCT4) and/or nuclear area (for gH2AX). While the images are important to include, they do not capture if the PRL3-induced PGCCs actually have higher staining, or whether the human eye only perceives it as such - namely, normalized by cell/nuclear area, is it equal to that of non-PGCCs?
5. The data presented in Fig 2b is somewhat confusing given the conclusions presented. It is stated throughout that PRL3 overexpression induces aberrant cell cycle, resulting in a single PGCC formation rather than 2 daughter cells. If this is true, the total cell number of CHO-PRL3 cells would not increase at the same rate as control. I am unsure if there is related data that I am missing, or if I am not following the conclusions of the paper as intended. In addition, the number

of PGCCs vs non-PGCCs in each group over the same time period would be useful.

6. Page 7, line 132-133: it is stated "hypothesized that recurrent tumors might be enriched with cells harboring extensive genome damage derived from parental PGCCs. To test this hypothesis, we analyzed the expression of PRL3 and gH2AX in four pairs...". While an interesting study, this western blot does not test the stated hypothesis. It tests the association with PRL3 and gH2AX expression, but does not show PGCC burden, nor that those tumors are "derived from parental PGCCs." In addition, a discussion of the lack of association in Pair3 is warranted.

7. How was the dose and timeframe of cisplatin exposure selected? If possible, a viability curve for the control population would be useful to include. The methods section states the dose at 5 ug/ml which is equal to ~16 uM, an extremely high dose (by way of reference, LD50 of MDA-MB-231 cancer cell line is only ~6 uM) that is not clinically meaningful. Do PGCCs form in the control population with lower cisplatin doses?

Is the drug spiked in daily, or was it a single dose? These details are important to include, especially given the statements about "prolonged chemotherapy" and nuclear area over time.

8. For Figure 3a and Fig 3b, what are the total cell counts / PGCC counts / non-PGCC cell counts at the presented time points? In lieu of these, inclusion of a viability/cell death curve would be valuable.

9. page 8 / figure 3, it is stated "some of these cells grew up to 100 times larger than untreated mononucleated cells." This data must be included.

10. page 8: "Survival of CHO-PRL3 cells correlated with increased nuclear size and cellular hypertrophy over the duration of a cisplatin treatment, resulting in the accumulation of a resistant PGCC population over time. These genomically-damaged cells escaped cell death and survived repeated cycles of cell division and endoreplication"

None of this data is included. If this statement remains in the manuscript, the following data should be included:

- The data demonstrating correlation of survival with increased nuclear size and with cellular hypertrophy must be presented.
- Data demonstrating that the resultant PGCCs are now resistant to a subsequent treatment of cisplatin must be shown.
- Data that the PGCCs underwent multiple cycles of cell division (this would result in non-polyloid cells) under treatment.
- Data that the PGCCs underwent endoreplication under treatment.

11. Page 8, lines 153-156 and Fig 3b: Fig 3b shows no significant difference in mean nuclear area. Are statistics missing from the figure? If not, the language should be revised.

12. Fig 3d – n=26 per group. Can the authors clarify what is meant here? Are these 26 separate biologic replicates?

13. Page 8 line 170 – Page 9 – "...difference was no longer apparent in the presence of ATM inhibition, which increased PGCC accumulation in both cell lines to similar levels." This data is not included in the current manuscript, therefore this statement and subsequent discussion should be removed (or data included).

14. In the mouse models, why is the mean tumor volume of the primary tumor so different in the treated and untreated groups? For B16F0: untreated at ~0.6 cm vs treated at ~0.25 cm (2.4 fold difference); For SNU-484: untreated at ~0.2 cm vs treated at ~0.05 cm (4 fold difference).

As it currently stands, it suggests that these mice were not adequately randomized. At a minimum, the Y axes should be on the same scale for Fig 4a/4b and for Fig 4c/4d.

While the data remain compelling, such a difference in starting tumor volume with untreated being much higher than the treated group does blunt the impact of the prevention of tumor relapse.

15. As mentioned above, it would be valuable to present the PGCC percentage of these cell line populations. On the basis of the first several figures, as both of these cell lines are PRL3 positive,

one would anticipate them to have high PGCC.

16. Figure 5 presents the nuclear area per cell of the relapsed tumors. More specific details regarding how these measurements were taken would be valuable. PGCC count from H&E is notoriously difficult as cell borders are difficult to discern. From the images shown, this reviewer is unable to define cell borders – how were they measured?

16. It would be valuable to see the same type of data from the few number of relapsed treated animals for the B16F10 model. Were the remaining cells also PGCCs?

17. Overall, the first section of the discussion is not supported by the presented data. Specifically, there is not data that show that PRL3-overexpression is sufficient to increase frequency of endoreplication, nor is it shown that PRL3-overexpression is sufficient to drive formation of PGCCs as there is no comparison to control.

In addition, the data do not specifically support that “PRL3+ PGCCs could be targeted successfully using PRL3-zumab.” While the mouse modeling show that PRL3-zumab reduces tumor regrowth, there is not evidence that it was specific targeting of PGCCs.

18. on page 14: “...our finding that PRL3-zumab can suppress the recurrence of PRL3+ hyperploid tumors...” This data was not shown. To make this statement, the relapsed tumors of the PRL3-zumab treated animals must be assessed.

19. on page 14: “...we sought to broadly characterize PRL3 protein expression in 151 hard-to-obtain, fresh-frozen patient tumor samples from 11 different cancer types...” What interesting, this data is not the focus of this paper, nor is it included in the present manuscript.

Minor concerns/revisions:

1. Page 6, line 106 – the video shown does not show “fusion.” If the cell fails to complete cytokinesis, there are not two separate cells, therefore they do not fuse. This is an important point to distinguish failed cytokinesis vs cell-cell fusion (another route of PGCC formation).

2. Throughout this section, the word “endoreplication” is used. The data presented do not prove endoreplication per se, only that cytokinesis is not completed. I would suggest revising the language to be more specific.

3. Throughout, single channel images should be included – especially for the color vision impaired, the IF images are difficult to interpret.

4. Page 6, lines 110-112 – it is unclear what is meant by “nuclear division” being “highly active” – without 3d imaging, it is not possible to confirm multinucleation or karyokinesis. Is it possible the authors intend to instead discuss DNA replication (therefore increase in ploidy) rather nuclear division?

5. page 9, line 183 – there is no data presented in the present manuscript that suggests that PGCCs are “dormant”

Reviewer #2:

Remarks to the Author:

Polyploid/multinucleated giant cancer cells (PGCCs) constitute a subset of cells within a solid tumor that is responsible for metastasis, therapy resistance and disease recurrence post-therapy. In this article, Thura et al. have presented compelling evidence demonstrating the key role played by PRL3 in mediating these events associated with PGCCs. In animal studies, the authors show that after tumor removal, the tumor relapses, as expected, and that the relapsed tumor is enriched with PGCCs; this relapse is shown to be reduced/prevented by treatment of animals with an

antibody drug against PRL3 (PRL3-zumab). It is concluded that PRL3 antibodies might be useful in adjuvant immunotherapy to target PRL3-expressing PGCCs and hence to prevent metastasis and relapse. The results are of high quality and support the conclusions. The following points need to be addressed to improve clarity.

(i) In the abstract it is stated that "PRL3-zumab...could reduce tumor relapse" whereas in the title and other places it is stated that PRL3-zumab prevents tumor relapse. The former implies that the treatment delays relapse, whereas the latter implies complete cure. This needs to be clarified. In Figure 1 Bii, the treatment does not seem to prevent relapse.

(ii) The first series of experiments involve CHO cells and not cancerous cells. Overexpression of PRL3 in these CHO cultures is shown to lead to enrichment of polyploid/multinucleated giant cells. The authors state that these giants are reminiscent to PGCCs (which is correct), but they also refer to the giant CHO cells as PGCCs (which is not necessarily correct). Thus, there is no evidence for the presence of PGCCs (polyploid giant CANCER cells) in Fig 1 (the giants are CHO giants and not necessarily giant cancer cells). The authors should consider referring to CHO giant cells as, e.g., polyploid/multinucleated giant CHO cells, but not PGCCs. Do the authors know if these giant CHO cells can promote tumors if injected in animals? If this information is not available, then to call them PGCCs could be misleading.

(iii) In figure 1, it is shown that one giant CHO cell is positive for SOX2, and one giant CHO cell is positive for OCT4. Do the authors know if all giant cells are positive for these factors, or only a subset is positive? This needs to be clarified. Also, image D seems to show OCT4 positive staining in all cells and not exclusively in the giant cell.

Reviewer #3:

None

Response to the reviewer's comments

Reviewer 1:

This manuscript has two main foci: the PRL3 and polyploid giant cancer cells (PGCCs) with a discussion of how expression of PRL3 may be related to PGCCs and how PRL3 may be related to recurrence. Utilizing constitutive overexpression in CHO cells, the authors show that PGCCs in the CHO cell line are PRL3 positive, and that those PGCCs express classical cancer stem-cell makers and are positive for gH2AX, a surrogate for double strand breaks. Next, evidence is presented that PGCCs are formed via failed cytokinesis. The link with chemotherapy resistance is made by presenting data that overexpressing PRL3 CHO cells (assumed to be PGCCs) have suppressed ATM signaling following long-term high-dose cisplatin treatment. In vivo data from two immunocompromised mouse models who that targeting PRL3 with PRL3-zumab reduces relapse following s.c. tumor removal; interestingly, data is presented that the "relapsed" tumors may be enriched for PGCCs. This data is consistent with prior published reports as well as presented H&E from patient tissue that recurrence and/or recurrence is enriched for PGCCs.

This paper is interesting and the data connecting PRL3 and PGCCs is new to the field, but the data presented do not support the conclusions or the title of the paper: specifically, there is not direct evidence that PRL3 induces PGCCs (necessarily a cancer cell phenomenon) nor that PRL3-zumab eliminates those PGCCs to prevent tumor relapse. Moreover, a number of necessary controls and necessary quantification are absent from the manuscript that limits interpretation of data. Separate from the lack of presented controls (presumably a simple revision to include), the data are sound. My recommendation is either to 1) revise the conclusions derived from these data to more accurately capture what data are presented or 2) include additional data to enable the reader (and this reviewer) to directly assess PRL3, PGCCs, and recurrence following chemotherapy.

Authors: Many thanks to the reviewer for your full endorsement of our manuscript. We are thankful for your hard efforts in reviewing and giving insightful comments on this article; we followed the first option as you suggested: *1) revise the conclusions derived from these data to more accurately capture what data are presented.* We have toned down several necessary statements for the conclusions to be coherent with the new title below:

PRL3 promotes induces Polyploid Giant Cancer Cells formation, while PRL3-zumab may eliminate PGCCs to reduce tumor relapse which are eliminated by PRL3-zumab to prevent tumor relapse

This is an important paper demonstrating PRL3 enhances PGCCs formation and PRL3-zumab could serve as an *Adjuvant Therapy* to inhibit tumor relapse. The efficacy of tumor relapse inhibition depends on the degree of tumor aggressiveness.

PRL3-zumab is currently in Phase 2 Clinical trials in Singapore¹, US², and China to treat multiple cancer types (Table 1):

Table 1: Web site links for clinical trials

Site	Clinical Trial Page
------	---------------------

Singapore ¹	https://clinicaltrials.gov/ct2/show/NCT04118114
USA ²	https://clinicaltrials.gov/ct2/show/NCT04452955

Regarding the reviewer's 25 questions (20 Specific points + 5 minor points), It will be very challenging to address all questions in this single article. Therefore, we will focus on two conclusions: '**PRL3 promotes PGCCs**' and '**PRL3-zumab could serve as Adjuvant Therapy**' addressed by below important points:

1. PRL3-zumab could be a potential *Adjuvant Therapy* after surgery, either to prevent or reduce cancer relapse depending on the tumor aggressiveness.
2. CHO-K1 cells with overexpressed PRL3 specific oncotarget showed large amount of PGCCs whereas B16F0 and SNU484 with lower endogenous PRL3 protein expression showed less PGCCs.
3. Enhanced PGCCs formation by PRL3 overexpression depends on PRL3 phosphates domain whereas PRL3 PTP mutant (C104S) abolishes PGCCs formation.
4. PRL3 overexpressing CHO cells are cancer cells.

The elucidation of the followings will be out of scope of this manuscript and will take more time to explore in our future independent research projects.

- (1) Molecular mechanism of how PRL3-PGCCs related to Stem cells,
- (2) Signaling pathway of PRL3 in PGCC formation
- (3) Basic molecular mechanism of how PRL3-PGCC was formed via cell cycle division details
- (4) Proof of evidence on how PRL3-zumab removes or kills PRL3-PGCCs for tumor relapse inhibition

In addition, since reviewer 2 supported our first submitted version with only 3 comments, we will maintain the original format focusing on data amendments related to the title.

Point 1. PRL3-zumab could be a potential Adjuvant Therapy after surgery, either to prevent or reduce cancer relapse depending on the tumor aggressiveness.

1.1 Animal models: We have reported that intravenous tail vein injection of CHO-K1 cells overexpressing GFP-PRL3 or overexpressing myc-PRL3 could induce metastatic lung tumors³ (Guo et al., Cancer Biology Therapy, 2008). Below is the Figure 2A from Guo et al showing CHO-PRL3 metastatic lung tumors.

Figure 2. PRL-3 and PRL-1 mAbs specifically inhibit the formations of their respective metastatic lung tumors. (A) 1×10^6 AT3 cells (described in Fig. 1) were injected into nude mice via the tail vein. Mice were either untreated (a, $n = 10$) or PBS-treated (b, $n = 10$), or treated with two unrelated antibodies (c, $n = 5$; d, $n = 5$). PRL-3 mAb 223 is in the form of purified IgG (e) or ascitic fluid (g); PRL-3 mAb 318 is in the form of purified IgG (f) or ascitic fluid (h) administrated via the tail vein. The different antibodies were injected on days 3, 6 and 9 post inoculation of AT3 cells. Lungs were dissected out on day 15 post-injection and photographed under fluorescence microscopy to show the GFP-positive metastatic tumors. Images a, b, e and f were lungs from female mice. Images c, d, g and h were lungs from male mice.

Although CHO-K1 PRL3 overexpressing cancer cells could form metastatic lung tumors by tail vein metastatic tumor models, the same cancer cells cannot form Xenograft subcutaneous tumors in ‘our tumor removal models’, therefore, we have to use different cell lines such as SNU484 (a human Gastric cancer cell line) or B16F0 (a more aggressive mouse melanoma cancer cell line), both are naturally occurring cancer cell lines expressing endogenous PRL3 protein. PRL3-zumab could completely (100%) inhibit SNU484 tumor relapse, however, for more aggressive B16F0 tumors, PRL3-zumab could only reduce >90% (but not 100%) of B16F0 tumor relapse. The efficacy of PRL3-zumab in inhibiting tumors relapse depending on the aggressiveness of the cancer types although the principle is the same that the tumors should express PRL3 oncotarget. PRL3-zumab could either prevent or reduce tumor relapse depending on the nature of tumors’ aggressiveness. To be conserve, we used ‘reduce’ (instead of ‘prevent’) in our new title for this article.

1.2 Antibody therapeutics: We generated PRL3 specific mouse antibody in 2005 (Li et al., Clinical Cancer Research 2005), chimeric antibody in 2007, then humanized antibody (PRL3-zumab) in 2012, screening and selecting more than 1 dozen cancer cell lines for PRL3 positive/negative cancer cell lines to be able to form tumor rapidly within our manageable timeframe (1-2 months) for tumor removal model with PRL3-zumab treatments. It took us >15 years from PRL3 gene identification, establishing stable cell pools below, to complete the project and the manuscript this far.

Point 2. CHO-K1 cells with overexpressed PRL3 specific oncotarget showed large amount of PGCCs whereas B16F0 and SNU484 with endogenously expressed PRL3 showed less PGCCs

In 1998, we identified PRL3 gene⁴ (Zeng et al., BBRC, 1998), and in 2001, Prof Vogelstein team from Johns Hopkins University reported that PRL3 was associated with cancer metastasis⁵ (Saha et al., Science, 2001). We then established CHO-K1 cells

overexpressing either myc-PRL3 (Figure 1A), or GFP-PRL3 (Figure 1B), high myc-PRL3 or GFP-PRL3 expressing cells showed in green fluorescent, nicely correlated with >95% of PGCC Phenomena, white arrows point at the polynuclear in DAPI blue stains at the middle panel. Within the same images, side by side with non-green cells, which were low or non-PRL3 expressing cells, serving as side by side internal controls on the same images, >95% of such cells were single nuclear, suggesting high GFP expressing cells and low GFP expressing cells perfectly match with polynuclear vs single nuclear. Consistently, we did not see much PGCCs in B16F0 ($2.6 \pm 0.8\%$) and SNU484 ($1.27 \pm 1.03\%$) cells, which are PRL3 positive naturally occurring cancer cell lines, yet the endogenous PRL3 expression is much less than exogenous overexpressing PRL3 of CHO-K1 cells.

We generated three CHO-K1 stable pools overexpressing 1. myc-PRL3, 2. GFP-PRL3 (wild-type) and 3. GFP-PRL3 (C104S), a PTP phosphatase mutation to correlate with PGCCs phenomena, and demonstrate PRL3 depends on its phosphatase activity to the phenomena of PGCCs. On the same images (Figure 1A and 1B), non-PRL3 (or low PRL3) expressing cells serve as internal controls side by side with high PRL3-expressing PGCCs. Overexpressing PRL3 levels are much higher than naturally occurring PRL3 positive cell lines (B16F0 and SNU484), which did not show much PGCCs, could be due to low PRL3 expression levels as compared with super high PRL3 overexpression in early transient transfection CHO-K1 cell lines. Moreover, these naturally occurring PRL3 positive cancer cells might have already adapted to endogenous PRL3 expression levels, and hence they did not show much PGCCs.

Point 3. Enhanced PGCC formation by PRL3 overexpression depend on PRL3 phosphates domain whereas PRL3PTP mutant (C104S) abolished PGCC formation.

Furthermore, in 2003⁶ (Zeng et al *Cancer Research*, 2003), we made point mutation at the PRL3 phosphatase active site position (C104S) to abolish PRL3 Phosphatase comparing with CHO-K1 cells expressing wildtype PRL3 vs PRL3 PTP mutant cells showed not only slower cell migration, but also reduced PGCCs cells: **Reference Video A**: CHO-cells overexpressing PRL3 wildtype (polynuclear) vs **Reference Video F**: CHO-cells overexpressing PRL3 PTP mutant (single nuclear), suggesting PRL3 derived PGCC formation depending on its phosphatase PTP domain. Immunofluorescence images below showed our conclusions that high PRL3 (in Green cells) correlated with PGCCs:

Reference Video A (CHO-PRL3)

CHO-PRL3

Reference Video F (CHO-PRL3-PTP mutant)

CHO-PRL3-PTP mutant

CHO- myc-PRL3 (green)

CHO-myc-PRL3 (Green)

CHO-GFP-PRL3 (green)

Point 4. PRL3 overexpressing CHO cells are cancer cells

Yes, either CHO-K1 overexpressing GFP-PRL3³ (Guo et al, Cancer Biology and Therapy, 2008) or overexpressing myc-PRL3, as shown below, are cancer cells⁶ (Zeng et al., Cancer Research, 2003), using CHO-K1 cells overexpressing myc- β -gal as controls to CHO-K1 cells overexpressing myc-PRL-3 and myc-PRL-1, CHO-K1 cells Promote Cell Migration, Invasion, and Metastasis⁶ (Below is Figure 6 of Zeng et al 2003).

Fig. 6. The expression of Myc-PRL-1 and Myc-PRL-3 promotes cell metastasis *in vivo*. Mice were examined 25 days after tail vein injection of (5 _ 105) Myc-PRL-1-, Myc-PRL-3-, or β -gal-expressing (clone13) cells. Metastatic tumors were found in the lungs of all of the mice injected with Myc-PRL-1 and Myc-PRL-3 cells, whereas 2 of the 10 mice injected with Myc-PRL-3 cells also developed liver metastases. The *top panels* show the gross morphology of the respective lungs and/or liver, whereas the *bottom panels* show the histological morphologies of sections derived from the respective tissues and stained with H&E. *T* stands for areas with tumor. Bars, 2.5 mm and 100 μ m for the *top* and *bottom* panels, respectively.

We hope this article will not be delayed further. Although we are unable to address all questions in this single article. We have tried our best to furnish **Point to Point** responses, modify several words/expressions used and tone down some conclusions drawn. We hope that the reviewers will find our revised version acceptable.

Point-by-point responses to Reviewer's comments

Revised words/ statements in manuscript were mentioned in blue color

Specific points are listed below:

1. It is curious that in this study of polyploid giant cancer cells that the non-cancer cell line model CHO was chosen. While a valuable model, it would be valuable to see the impact of PRL3 overexpression in a cancer cell line model. This is especially important given the discordance between the overexpression PRL CHO cells showing a high level of PGCCs (e.g., Fig 1a) and the low level (and perhaps absence of) PGCCs in the PRL+ F16F0 and SNU-484 primary tumors (Fig 5ai and bi). This is especially critical given the conclusion that PRL3 expression alone is sufficient to induce PGCC formation.

Authors: Many thanks to the Reviewer for his/her insightful comment. CHO-K1 is a normal cell line, however when PRL3 is overexpressed in CHO-K1, it becomes cancer cells, which we confirmed **by metastatic tail vein model**^{3,6}. These CHO-PRL3 cells can metastasize to lung and liver tissues^{3,6} (Zeng et al *Cancer Research* 2003).

Although the same CHO-PRL3 cells can induce tumors to lung and liver by metastatic tail vein model, it cannot form tumors in subcutaneous tumor model to establish the tumor removal model. Therefore, we have to use two naturally occurring PRL3 positive cancer cell lines: 1. SNU484, human gastric cancer cell line and 2. B16F0, mouse melanoma cell line, both cancer cell lines can be used to establish tumor removal model. Here we found that significantly larger mean nuclear size in relapse tumor (Fig 5a iii, 5b

iii) indicating relapse tumor might be formed by abundant of PGCCs. PRL3-zumab could block completely (100%) SNU484 tumor relapse. However, B16F0 is a very aggressive cancer cell line, PRL3-zumab could not completely block, but could reduce >90% of tumor relapse, suggesting that the inhibition efficacy of tumor relapse depends on the aggressiveness of the cancer type, but the principle is the same that the tumors should express PRL3 oncotarget.

2. Appropriate negative controls for CHO-myc-PRL3 and CHO-PRL3 are absent throughout. Non-manipulated CHO cell line is not an appropriate control as PGCCs are well known to form upon transfection or lentiviral transduction in cell culture. A control such as the same promoter driving a marker would be more appropriate - this reviewer is unsure if that is the same as the “CHO-GFP” presented in Fig 2b and Fig2c. Images of such the control cell line should be included in Fig 1a, Fig 1b, Fig 1c, Fig 1d, Fig 2d, Fig 3a, Fig 3b, Fig 3c and Fig 3d.

Authors: We thank the reviewer’s comments. Within the same images (Figure 1A and 1B), low or non-expressing myc-PRL3 and GFP-PRL3 cells served as internal controls side by side with highly myc-PRL3 and GFP-PRL3 expressing cells on the same images, **ReferenceVideo F** (cited in point 3) showed PRL3 PTP mutant abolished PGCCs.

Here, as the reviewer suggested, we used CHO-PRL3 (C104S) PTP mutant, here in we called CHO-GFP as negative control for CHO-GFP-PRL3(CHO-PRL3) cells to compare the formation of PGCCs. The control used in Figure 3 are CHO-GFP cells. Below are the images of CHO-GFP and CHO-PRL3. Abundance of PGCCs can be seen in CHO-PRL3 cells. We added this image as Supplementary Figure 1a.

Supplementary Figure 1a: PRL3 overexpression causes the formation of polyploid giant cells.

(a) Immunofluorescence analysis of CHO cell lines stably expressing EGFP-tagged PRL3 (CHO-PRL3) and EGFP-PRL3-C104S mutant (CHO-GFP). Multinucleated polyploid giant cells can be seen abundantly in CHO-PRL3 cells. *Bar*, 20 μ m

3. A number of compelling images are presented throughout the manuscript, but quantification is lacking. For example, in Fig 1, PGCCs are shown via IF. What is the % PGCC of the population? What is the total number? Is it greater than control? Without this data, it is impossible to determine if PRL3 impacts PGCC formation or survival.

Authors: We thank the reviewer for pointing out the important facts. As suggested, we did the quantification of the IF images, comparing the % of PGCC in CHO-GFP cells vs CHO-PRL3 cells. The results indicated that the % of PGCC is $6.7 \pm 4\%$ in CHO-GFP cells and $35.8 \pm 12\%$ in CHO-PRL3 cells, indicating that the % of PGCC is >5 times increased by PRL3 overexpression. We have added this data in line 89 of manuscript and add the figure as Supplementary Figure 1b.

Supplementary Figure 1b. PRL3 overexpression causes the formation of polyploid giant cells.

Quantitative analysis of Polyploid giant cells in CHO-GFP vs CHO-PRL3. % of PGCC was significantly higher in CHO-PRL3 cells ($6.7 \pm 4\%$ in CHO-GFP vs $35.8 \pm 12\%$ in CHO-PRL3). Data obtained from cell counts of 3 biological replicates (3 fields in each replicates). $P < 0.0001$ (Student's *t*-test).

4. Similarly, it would be useful to include quantification of staining (e.g., PRL3, SOX2, OCT4, gH2AX), including normalization by cell area (for PRL3, SOX2, OCT4) and/or nuclear area (for gH2AX). While the images are important to include, they do not capture if the PRL3-induced PGCCs actually have higher staining, or whether the human eye only perceives it as such – namely, normalized by cell/nuclear area, is it equal to that of non-PGCCs?

Authors: Thank you for the comment. We will explore the molecular mechanism of how PRL3-PGCCs relates to Stem cell like markers in great details in the future research. In this article, we only showed a preliminary link between PGCCs and stem cell markers. As the reviewer suggested, we did the fluorescence intensity quantification for the staining of PRL3, SOX2, OCT4 and gH2AX using image J software. We analyzed the fluorescence intensity in CHO-PRL3-PGCC, CHO-PRL3 cells and CHO cells (which do not express PRL3) of the same images.

Quantification of fluorescent intensity indicated that PRL3, SOX2, OCT4 and gH2AX expression are highest in PGCCs compared to mononucleated cells. We have added these result in line 94 & line 123 of manuscript and added the figure as Supplementary Figure 2 & 3.

Supplementary Figure 2. SOX2 and OCT4 are highly expressed in CHO-PRL3-PGCC cells.

(a)

(b)

(c)

(a) Fluorescence Intensity of PRL3 and SOX2 was analyzed in nuclear and cytoplasmic area of CHO-PRL3-PGCC cells, CHO-PRL3 cells and CHO cells of same images. Both PRL3 & SOX2 expression are significantly highest in PGCC cells. SOX2 is higher in nuclear area of all 3 cell types. $P < 0.0001$ (Unpaired t test) for PRL3 expression. $P < 0.0001$ (One-way ANOVA) for SOX2 expression. (b, c) Fluorescence Intensity of PRL3 (cytoplasmic) and OCT4 (nuclear) were analyzed in CHO-PRL3-PGCC cells, CHO-PRL3 cells and CHO cells of same images. (b) Both PRL3 & OCT4 expression were highest in CHO-PRL3-PGCC cells indicating high PRL3 expression is associated with high OCT4 expression. (c) Fluorescence intensity of OCT4 from (b).

P<0.0001 (Unpaired t test) for PRL3 expression. P<0.0001 (One-way ANOVA) for OCT4 expression. Data represent mean ± SD from 2 biological replicates.

Supplementary Figure 3. gH2AX is highly expressed in CHO-PRL3- PGCC cells.

Fluorescence Intensity of PRL3 (cytoplasmic) and gH2AX (nuclear) were analyzed in CHO-PRL3-PGCC cells, CHO-PRL3 cells and CHO cells of same images. (a) Both PRL3 & gH2AX expression were highest in CHO-PRL3-PGCC cells indicating high PRL3 expression is associated with high gH2AX expression. (b) Fluorescence intensity of OCT4 from (a). P<0.0001 (Student's *t*-test) for PRL3 expression. P<0.0001 (One-way ANOVA) for gH2AX expression. Data represent mean ± SD from 3 biological replicates.

5. The data presented in Fig 2b is somewhat confusing given the conclusions presented. It is started throughout that PRL3 overexpression induces aberrant cell cycle, resulting in a single PGCC formation rather than 2 daughter cells. If this is true, the total cell number of CHO-PRL3 cells would not increase at the same rate as control. I am unsure if there is related data that I am missing, or if I am not following the conclusions of the paper as intended. In addition, the number of PGCCs vs non-PGCCs in each group over the same time period would be useful.

Authors: We agree with the reviewer that these are very important topics to explore whether PGCCs are defected in cell cycles or are they caused by rapidly uncontrolled DNAs replications, we requested to be our future research areas: Signaling pathway of

PRL3 in PGCCs formation and Basic molecular mechanism of how PRL3-PGCCs are formed via cell cycle division details.

In Fig 2b, proliferation rate of CHO-GFP versus CHO-PRL3 cells, was measured under normal condition where we detected that the proliferation rate of CHO-PRL3 was nearly the same with control, CHO-GFP.

PRL3 was reported to enhance cell migration, proliferation, metastasis and PGCC formation. Under normal conditions, CHO-PRL3 tends to proliferate faster than CHO-GFP. The faster proliferation rate in mononuclear CHO-PRL3 might compensate for the total cells and hence no differences in overall cell count.

We count the cells in each time point as total cell count and not as PGCCs vs non-PGCCs cells. So, we seek the reviewer's understanding for not having the data of PGCCs vs non-PGCCs in each group over the same time period.

6. Page 7, line 132-133: it is stated "hypothesized that recurrent tumors might be enriched with cells harboring extensive genome damage derived from parental PGCCs. To test this hypothesis, we analyzed the expression of PRL3 and γ H2AX in four pairs...". While an interesting study, this western blot does not test the stated hypothesis. It tests the association with PRL3 and γ H2AX expression, but does not show PGCC burden, nor that those tumors are "derived from parental PGCCs." In addition, a discussion of the lack of association in Pair3 is warranted.

Authors: We appreciate the reviewer's insightful comment. We did check the PGCCs burden in these tumor pairs by using Haematoxylin & Eosin (H&E) stain which was presented in Fig 5C. PGCC dominant cells were detected in Metastatic and recurrent tumors.

Here, we tested checkpoint protein, pChk2 in the western blot of these tumor pairs. PRL3 upregulation was associated with γ H2AX and pCHK2 upregulation.

Although γ H2AX was detected in primary tumor and pChk2 was not detected in recurrent tumor of pair 4, we considered this might be the nature of individual tumor because not like other tumor pair 1-3, pair 4 is primary and recurrence tumor, not metastasis, where the nature of recurrent tumor might be the same as primary tumor specifically for this pair only and does not follow our hypothesis.

We rewrite the manuscript as below in Page 7 & 8 (line 134 to line 160) of our manuscript (Blue color indicated newly added words) and revised the Figure 2f by adding pChk2 western blot.

"Unrepaired DSBs induce genome instability and promote tumorigenesis. One of the most important kinases activating cell cycle checkpoints following DNA damage is ATM (ataxia teleangiectasia mutated). ATM is a major physiological mediator of H2AX phosphorylation in response to DSB formation, and phosphorylates many other substrates including the checkpoint proteins, Chk1 and Chk2^{33, 34}. CHK2 is phosphorylated on the priming site T68 by ATM³⁵.

Since PGCCs have been reported to demonstrate chemo-resistance in human cancers and can give rise to therapy-resistant progeny with enhanced tumorigenicity^{26, 36}, we hypothesized that recurrent tumors might be enriched with cells harboring extensive genome damage derived from parental PGCCs. To test this hypothesis, we analyzed the expression of PRL3 and γ H2AX in four pairs of post-chemotherapy primary tumors and patient-matched metastatic or recurrent tumors; Pair 1: breast invasive carcinoma sample,

and its metastatic tumor in the brain; pair 2: colon adenocarcinoma sample and its metastatic tumor in the liver; pair 3: rectal cancer sample and its metastatic tumor in the lung and pair 4: thyroid cancer sample and its recurrent thyroid tumor. PRL3 expression was detected in all metastatic/ recurrent tumors and primary tumors of pairs 3 & 4. Importantly, markedly elevated expression of γ H2AX seen in metastatic/ recurrence tumor samples of pairs 1 to 4, but not detected in primary tumors apart from primary tumor of pair 4. Elevation of Phosphorylated Chk2 (pChk2) was also detected in metastatic tumor of pair 1 to 3 but not detected in recurrent tumor of pair 4. Both γ H2AX and pCHK2 were upregulated in metastatic tumor of pairs 1 to 3 but not detected in primary tumor. Additionally, the expression were also correlated with PRL3 expression (Figure 2f).

Pairs 1 to 3 are primary and metastatic tumors whereas pair 4 is primary and recurrent tumor. The nature of recurrent tumor might be the same as primary tumor specifically for this pair 4 only and does not follow our hypothesis, expressing γ H2AX in primary tumor and not expressing pChk2 in recurrent tumor.

These results support our *in vitro* observations, and suggest that metastatic/ recurrent tumors, which have upregulated PRL3 expression, and are able to survive and resist cell death despite extensive DNA damage.”

Previous Figure 2f

Updated Figure 2f

7. How was the dose and timeframe of cisplatin exposure selected? If possible, a viability curve for the control population would be useful to include. The methods section states the dose at 5 ug/ml which is equal to ~16 uM, an extremely high dose (by way of reference, LD50 of MDA-MB-231 cancer cell line is only ~6 uM) that is not clinically meaningful. Do PGCCs form in the control population with lower cisplatin doses? Is the drug spiked in daily, or was it a single dose? These details are important to include,

especially given the statements about “prolonged chemotherapy” and nuclear area over time.

Authors: We apologize for our unclear explanation. For the dose selection, we did the preliminary study using 1µg/mL, 5µg/mL and 10µg/mL of cisplatin and choose 5µg/mL as it can give optimum result.

PGCCs can form in control population until Day 4 days but does not survive and it is reduced until nearly absent in Day 7 and totally disappear in Day 11.

The cell culture media was changed every 48 hours together with the drug.

We revised Method section of our manuscript by adding the above information.

8. For Figure 3a and Fig 3b, what are the total cell counts / PGCC counts / non-PGCC cell counts at the presented time points? In lieu of these, inclusion of a viability/cell death curve would be valuable.

Authors: Please see below table for the total cell counts/ PGCCs counts and non-PGCCs cells count at the presented time points. We counted PGCCs vs non-PGCCs in Days 7 and 10 when we can see obvious differences. We counted the fixed cells on glass slides. So, we could not produce viability/ cell death curve. We seek the reviewer’s understanding for the missing data.

		Day 0	Day 4	Day 7	Day 11
CHO-GFP	Total cell count	4455	3391	32	49
	PGCC count	–	–	6	0
	non PGCC count	–	–	26	49
CHO-PRL3	Total cell count	5022	1850	3106	1297
	PGCC count	–	–	1647	683
	non PGCC count	–	–	1459	614

9. page 8 / figure 3, it is stated “some of these cells grew up to 100 times larger than untreated mononucleated cells.” This data must be included.

Authors: We apologize for our incomplete presentation. Below is the table showing mean cell area of one of the cisplatin treated CHO-PRL3-PGCCs. The cells grew up to multiple times much larger (as below) than untreated mononucleated cells. We added that data as Supplementary Figure 4. We used “multiple time” (Line 168) instead of 100 times in Page 8 of our manuscript. The figure of images and area of cells in different time point after cisplatin treatment was added as Supplementary figure 4.

Supplementary Figure 4. Cellular area of CHO-PRL3 cells after cisplatin treatment.

(a)

(b)

	Mean Area (μm^2) of CHO-PRL3 cells		
	Untreated	Cisplatin treated PGCC	
		Day 7	Day 11
Mean	312.90	29676.27	31943.27
S.D	115.01	11751.79	22277.59

(a) Immunofluorescence image of CHO-PRL3 mononuclear cells at Day 0 without any treatment and PGCCs of Day 7 and Day 11 after cisplatin treatment. Bar 50 μM . (b) Mean cellular area with standard deviation for CHO-PRL3 mononucleated untreated cells and cisplatin treated Day 7 & Day 11 CHO-PGCC-PRL3 cells.

10. page 8: “Survival of CHO-PRL3 cells correlated with increased nuclear size and cellular hypertrophy over the duration of a cisplatin treatment, resulting in the accumulation of a resistant PGCC population over time. These genomically-damaged cells escaped cell death and survived repeated cycles of cell division and endoreplication” None of this data is included. If this statement remains in the manuscript, the following data should be included:

- The data demonstrating correlation of survival with increased nuclear size and with cellular hypertrophy must be presented.
- Data demonstrating that the resultant PGCCs are now resistant to a subsequent treatment of cisplatin must be shown.
- Data that the PGCCs underwent multiple cycles of cell division (this would result in non-polyploid cells) under treatment.
- Data that the PGCCs underwent endoreplication under treatment.

Authors: We thank the reviewer for pointing out the required data for our statement. As our data is incomplete, we removed our statement from our manuscript.

11. Page 8, lines 153-156 and Fig 3b: Fig 3b shows no significant difference in mean nuclear area. Are statistics missing from the figure? If not, the language should be revised.

Authors: We thank the reviewer for pointing out the missing facts. The statistics of Fig3b is $P=0.05$ (two-way ANOVA). $P<0.0001$ CHO-PRL3 ($414 \pm 292 \mu\text{m}^2$) vs Ctrl (113 ± 51)

um³) (t test). The data was added in Line 170 & 171 of manuscript and figure legend of Figure 3.

12. *Fig 3d – n=26 per group. Can the authors clarify what is meant here? Are these 26 separate biologic replicates?*

Authors: Sorry for the unclear statement. These are technical replicates, different views on the same slide. We have made sure that there was no overlap between each FOV (i.e. each cell was never counted twice).

13. *Page 8 line 170 – Page 9 – “...difference was no longer apparent in the presence of ATM inhibition, which increased PGCC accumulation in both cell lines to similar levels.” This data is not included in the current manuscript, therefore this statement and subsequent discussion should be removed (or data included).*

Authors: Sorry for our irrelevant statement. As the reviewer suggested, we have removed that statement from our manuscript.

14. *In the mouse models, why is the mean tumor volume of the primary tumor so different in the treated and untreated groups? For B16F0: untreated at ~0.6 cm vs treated at ~0.25 cm (2.4 fold difference); For SNU-484: untreated at ~0.2 cm vs treated at ~0.05 cm (4 fold difference).*

As it currently stands, it suggests that these mice were not adequately randomized. At a minimum, the Y axes should be on the same scale for Fig 4a/4b and for Fig 4c/4d. While the data remain compelling, such a difference in starting tumor volume with untreated being much higher than the treated group does blunt the impact of the prevention of tumor relapse.

Authors: We agree with the reviewer’s comment. Actually these mice were adequately randomized. The tumor size of primary tumor in treated and untreated groups of all B16F0 & SNU-484 are nearly equal. That can be seen from images of animal and tumors in Fig 4ai, 4bi and 4ci, 4di.

There are differences in the figures because of our mistake in the scale of y axis of Fig 4b iii & d iii) during the preparation of figures. We apologize for our mistake and seek the reviewer’s understanding. We have prepared thoroughly and updated tumor size in pages 10, 11 and Figure 4 in manuscript.

15. *As mentioned above, it would be valuable to present the PGCC percentage of these cell line populations. On the basis of the first several figures, as both of these cell lines are PRL3 positive, one would anticipate them to have high PGCC.*

Authors: We understand the reviewer’s concern of presenting PGCC% of B16F0 and SNU 484 population. As these cell lines are naturally occurring PRL3 positive cancer cells, PGCC % is low compared to PRL3 overexpressed CHO cell line. In the *in vitro* cell culture in normal condition, % of PGCCs is $2.6 \pm 0.8\%$ in B16F0 cells and $1.27 \pm 1.03\%$ in SNU 484 tumor (shown in table below). We added the data in Line 204 & 218 of manuscript and added the figure as supplementary figure 5. Although we did not find much PGCCs in culture, we found higher% of PGCCs in relapsed ‘untreated’ tumor

sections, suggesting tumor microenvironments may induce PGCCs more effectively than *in vitro* cell culture.

Supplementary Figure 5. % of PGCC in B16F0 and SNU-484 in cell culture

(a)

(b)

(a) Immunofluorescence images of B16F0 and SNU484 cells culture in normal condition. Green color represents PRL3. White arrow indicates PGCCs. Bar 10 μ M. (b) Quantitative analysis of Polyploid giant cells in B16F0 and SNU-484 cells. CHO-PRL3. % of PGCCs

was $2.6 \pm 0.8\%$ in B16F0 cells and $1.27 \pm 1.03\%$ in SNU-484 cells. Data obtained from cell counts of 3 biological replicates (3 fields in each duplicate).

16. *Figure 5 presents the nuclear area per cell of the relapsed tumors. More specific details regarding how these measurements were taken would be valuable. PGCC count from H&E is notoriously difficult as cell borders are difficult to discern. From the images shown, this reviewer is unable to define cell borders – how were they measured?*

Authors: We agree with the reviewer's comment that it is difficult to discern cell border in H&E stain images. Here, we measure nuclear area only and compare the mean nuclear area of all the cells in slides obtained from primary and relapse tumor in Figure 5a and b. We corrected the labels in Figure 5a iii and b iii as "Mean nuclear area". Abundance of PGCCs can be seen in Figure 5a ii, b ii as cells with enlarged nuclei.

17. *It would be valuable to see the same type of data from the few number of relapsed treated animals for the B16F10 model. Were the remaining cells also PGCCs?*

Authors: As the size of relapse treated tumor in B16F0 model was very small and scanty, we could not manage to analyze it. We apologize for not presenting the data and seek his/her understanding.

18. *Overall, the first section of the discussion is not supported by the presented data. Specifically, there is not data that show that PRL3-overexpression is sufficient to increase frequency of endoreplication, nor is it shown that PRL3-overexpression is sufficient to drive formation of PGCCs as there is no comparison to control.*

Authors: We thank the reviewer for his/her suggestion. We have replaced "endoreplication" with "incomplete cytokinesis" caused by PRL3 overexpression in the manuscript. As we answered in **Point 3**, we added the video comparing cell growth of CHO-PRL3 and CHO-PRL3 mutant together with IF images of CHO-myc-PRL3 and CHO-PRL3.

In addition, the data do not specifically support that "PRL3+ PGCCs could be targeted successfully using PRL3-zumab." While the mouse modeling show that PRL3-zumab reduces tumor regrowth, there is not evidence that it was specific targeting of PGCCs.

Authors: We detected that PRL3+B16F0 and SNU484 recurrent tumors were enriched with PGCC-like cells with enlarged nuclei. With the treatment of PRL3-zumab, there is no recurrence in SNU-484 tumors and only <10% recurrence in B16F0 tumor and we considered PRL3-zumab inhibited PGCC-like recurrent tumors. We seek the reviewer's understanding for our statement.

19. *on page 14: "...our finding that PRL3-zumab can suppress the recurrence of PRL3+ hyperploid tumors..." This data was not shown. To make this statement, the relapsed tumors of the PRL3-zumab treated animals must be assessed.*

Authors: We thank the reviewer for the comment. As we presented in comment 18, we detected that PRL3+B16F0 and SNU484 recurrent tumors were enriched with PGCC-like cells with enlarged nuclei. With the treatment of PRL3-zumab, there is no recurrence in SNU-484 tumors and only <10% recurrence in B16F0 tumor and we considered PRL3-zumab inhibited PGCC-like recurrent tumors since we found higher % of PGCCs in relapsed tumor sections from ‘untreated’ than ‘treated’ mice.

We have rephrased and softened the tone of our words in line 294, 295 (Page 14) of the manuscript.

20. *on page 14: “...we sought to broadly characterize PRL3 protein expression in 151 hard-to-obtain, fresh-frozen patient tumor samples from 11 different cancer types...” What interesting, this data is not the focus of this paper, nor is it included in the present manuscript.*

Authors: We apologize for our unclear presentation. We cited this data from our previous study⁷ (Thura et al., Nature Communications 2019). We have rewritten the manuscript and cited the reference accordingly.

Minor concerns/revisions:

1. *Page 6, line 106 – the video shown does not show “fusion.” If the cell fails to complete cytokinesis, there are not two separate cells, therefore they do not fuse. This is an important point to distinguish failed cytokinesis vs cell-cell fusion (another route of PGCC formation).*

Authors: We thank the reviewer for highlighting the important point. We have removed “fusion” from the manuscript.

2. *Throughout this section, the word “endoreplication” is used. The data presented do not prove endoreplication per se, only that cytokinesis is not completed. I would suggest revising the language to be more specific.*

Authors: We agree ‘endoreplication’ is not a proper word. We replaced it with ‘incomplete cytokinesis’ in the manuscript.

3. *Throughout, single channel images should be included – especially for the color vision impaired, the IF images are difficult to interpret.*

Authors: As the reviewer advised, we added single channel images in Figure 1 (a, b, c, d), Supplementary Figure 1& 5.

4. *Page 6, lines 110-112 – it is unclear what is meant by “nuclear division” being “highly active” – without 3d imaging, it is not possible to confirm multinucleation or karyokinesis. Is it possible the authors intend to instead discuss DNA replication (therefore increase in ploidy) rather nuclear division?*

Authors: Thanks for the comment. We have corrected the manuscript by removing nuclear division and discussing uncoupling DNA replication promotes ploidy.

5. page 9, line 183 – there is no data presented in the present manuscript that suggests that PGCCs are “dormant”

Authors: We agree and have removed the word “dormant” in manuscript.

References:

1. Phase II Study of PRL3-ZUMAB in Advanced Solid Tumors, NCT04118114, U.S. National Library of Medicine, *ClinicalTrials.gov*, <https://www.clinicaltrials.gov/ct2/show/NCT04118114>
2. A Study to Assess Safety and Efficacy of PRL3-Zumab in Patients With Solid Tumors, NCT04452955, U.S. National Library of Medicine, *ClinicalTrials.gov*, <https://clinicaltrials.gov/ct2/show/NCT04452955>.
3. Guo K, Tang JP, Li J, Zeng Q, Monoclonal antibodies target intracellular PRL phosphatases to inhibit cancer metastases in mice. *Cancer Biology & Therapy*. **7(5)**, 750-757 (2008).
4. Zeng Q, Hong W, Tan YH, Mouse PRL2 and PRL3, two potentially prenylated protein tyrosine phosphatases homologous to PRL1. *Biochem Biophys Res Commun*, **244**, 421-427 (1998).
5. Saha S, Bardelli A, Buckhaults P, Velculescu VE, Rago C, St Croix B, Romans KE, Choti MA, Lengauer C, Kinzler KW, Vogelstein B, A phosphatase associated with metastasis of colorectal cancer. *Science*. **294**,1343-46 (2001)
6. Zeng Q, Dong JM, Guo K, Li J, Tan HX, Koh V, PRL-3 and PRL-1 promote cell migration, invasion, and metastasis. *Cancer Res*. **63**, 2716-22 (2003).
7. Thura M, Al-Aidaros AQ, Gupta A, Chee CE, Lee SC, Hui KM, Li J, Guan YK, Yong WP, SO J, Chng WJ, Ng CH, Zhou NJ, Wang LZ, Yuen JSP, Ho HSS, Yi SM, Chiong E, Choo SP, Ngeow J, Ng MCH, Chua C, Yeo ESA, Tan IH, Sng JXE, Tan NYZ, Thierry JP, Goh BC, Zeng Q, PRL3-zumab as an immunotherapy to inhibit tumors expressing PRL3 oncoprotein. *Nature communications*. **10**,2484 (2019).

Reviewer #2:

Polyploid/multinucleated giant cancer cells (PGCCs) constitute a subset of cells within a solid tumor that is responsible for metastasis, therapy resistance and disease recurrence post-therapy. In this article, Thura et al. have presented compelling evidence demonstrating the key role played by PRL3 in mediating these events associated with PGCCs. In animal studies, the authors show that after tumor removal, the tumor relapses, as expected, and that the relapsed tumor is enriched with PGCCs; this relapse is shown to be reduced/prevented by treatment of animals with an antibody drug against PRL3 (PRL3-zumab). It is concluded that PRL3 antibodies might be useful in adjuvant immunotherapy to target PRL3-expressing PGCCs and hence to prevent metastasis and relapse. The results are of high quality and support the conclusions. The following points need to be addressed to improve clarity.

Authors: Many thanks for your strong support with kind encouraging compliment that ‘The results are of high quality and support the conclusions.’

Point-by-point responses to Reviewer’s comments

Revised words/ statements in manuscript were mentioned in blue color

(i) *In the abstract it is stated that “PRL3-zumab...could reduce tumor relapse” whereas in the title and other places it is stated that PRL3-zumab prevents tumor relapse. The former implies that the treatment delays relapse, whereas the latter implies complete cure. This needs to be clarified. In Figure 1 Bii, the treatment does not seem to prevent relapse.*

Authors: Many thanks for this important point.

We have toned down several statements for the conclusions to be coherent with the new title. PRL3-zumab could either prevent or reduce tumor relapse depending on the nature of tumors’ aggressiveness. To be conservative, we used ‘reduce’ (instead of ‘prevent’) in our new title for this article.

PRL3 promotes induces Polyploid Giant Cancer Cells formation, while PRL3-zumab may eliminate PGCCs to reduce tumor relapse which are eliminated by PRL3-zumab to prevent tumor relapse

(ii) *The first series of experiments involve CHO cells and not cancerous cells. Overexpression of PRL3 in these CHO cultures is shown to lead to enrichment of polyploid/multinucleated giant cells. The authors state that these giants are reminiscent to PGCCs (which is correct), but they also refer to the giant CHO cells as PGCCs (which is not necessarily correct). Thus, there is no evidence for the presence of PGCCs (polyploid giant CANCER cells) in Fig 1 (the giants are CHO giants and not necessarily giant cancer cells). The authors should consider referring to CHO giant cells as, e.g., polyploid/multinucleated giant CHO cells, but not PGCCs. Do the authors know if these giant CHO cells can promote tumors if injected in animals? If this information is not available, then to call them PGCCs could be misleading.*

Authors: We are thankful to the reviewer for this important question: CHO-K1 is a normal cell line, however, either myc-tag-PRL3 or GFP-tag-PRL3 overexpressed in CHO-K1 will cause cancerous cells. Please see our previous publication, using control cells myc- β -gal CHO-K1 to demonstrate: myc-PRL-3 and myc-PRL-1 CHO-K1 cells Promote Cell Migration, Invasion, and Metastasis (Figure 6)¹ (Zeng et al Cancer Research 2003)

Fig. 6. The expression of Myc-PRL-1 and Myc-PRL-3 promotes cell metastasis *in vivo*. Mice were examined 25 days after tail vein injection of (5 ± 105) Myc-PRL-1-, Myc-PRL-3-, or β -gal-expressing (clone13) cells. Metastatic tumors were found in the lungs of all of the mice injected with Myc-PRL-1 and Myc-PRL-3 cells, whereas 2 of the 10 mice injected with Myc-PRL-3 cells also developed liver metastases. The *top panels* show the gross morphology of the respective lungs and/or liver, whereas the *bottom panels* show the histological morphologies of sections derived from the respective tissues and stained with H&E. *T* stands for areas with tumor. Bars, 2.5 mm and 100 μ m for the *top* and *bottom* panels, respectively.

or CHO-GFP-PRL3 cancer cells² (Guo et al Cancer Biology and Therapy2008):

Figure 2. PRL-3 and PRL-1 mAbs specifically inhibit the formations of their respective metastatic lung tumors. (A) 1×10^6 AT3 cells (described in Fig. 1) were injected into nude mice via the tail vein. Mice were either untreated (a, n = 10) or PBS-treated (b, n = 10), or treated with two unrelated antibodies (c, n = 5; d, n = 5). PRL-3 mAb 223 is in the form of purified IgG (e) or ascitic fluid (g); PRL-3 mAb 318 is in the form of purified IgG (f) or ascitic fluid (h) administrated via the tail vein. The different antibodies were injected on days 3, 6 and 9 post inoculation of AT3 cells. Lungs were dissected out on day 15 post-injection and photographed under fluorescence microscopy to show the GFP-positive metastatic tumors. Images a, b, e and f were lungs from female mice. Images c, d, g and h were lungs from male mice.

Although the same myc-PRL3 CHO-K1 or GFP-PRL3-CHO-K1 can induce metastasis tumors to lung and liver by tail vein, they cannot form Xenograft local tumors in ‘our tumor removal models’, therefore, we used two naturally occurring PRL3 positive cancer cell lines: 1. SNU484, human gastric cancer cell line and 2. B16F0, mouse melanoma cell line, both cancer cell lines can be used to establish tumor removal model. PRL3-zumab could block completely (100%) SNU484 tumor relapse. However, B16F0 is a very aggressive cancer cell line, PRL3-zumab could not completely block, but could reduce >90% of tumor relapse, suggesting that the inhibition efficacy of tumor relapse depends

on the aggressiveness of the cancer types, but the principle is the same that the tumors should express PRL3 oncotarget.

(iii) In figure 1, it is shown that one giant CHO cell is positive for SOX2, and one giant CHO cell is positive for OCT4. Do the authors know if all giant cells are positive for these factors, or only a subset is positive? This needs to be clarified. Also, image D seems to show OCT4 positive staining in all cells and not exclusively in the giant cell.

Authors: We really appreciate you are a supportive reviewer to this project. ‘PGCCs correlated with Stem –like cells’ would be a great topic for our future detailed investigation. In this short article, we mainly focus on two conclusions: ‘PRL3 promotes PGCCs’ and ‘PRL3-zumab could serve as Adjuvant Therapy’. SOX2 and OCT4 are positive in a subset (but not all) of PGCCs giant cells that are highly overexpressing PRL3 oncoprotein.

To clarify the expression of these markers in PGCC, we did fluorescence intensity quantification for staining of PRL3, SOX2, OCT4 and gH2AX using image J software. We analyzed the fluorescence intensity in CHO-PRL3-PGCC, CHO-PRL3 cells and CHO cells (which does not express PRL3) of the same images.

Quantification of fluorescent intensity indicated PRL3, SOX2, OCT4 and gH2AX expression are highest in PGCCs compared to mononucleated cells.

We have added these result in line 94 & line 123 of manuscript and added the figures as Supplementary Figure 2 & 3 in our manuscript.

Supplementary Figure 2. SOX2 and OCT4 are is highly expressed in CHO-PRL3-PGCC cells.

(a)

(b)

(c)

(a) Fluorescence Intensity of PRL3 and SOX2 was analyzed in nuclear and cytoplasmic area of CHO-PRL3-PGCC cells, CHO-PRL3 cells and CHO cells of same images. Both PRL3 & SOX2 expression are significantly highest in PGCC cells. SOX2 is higher in nuclear area of all 3 cell types. $P < 0.0001$ (Unpaired t test) for PRL3 expression. $P < 0.0001$ (One-way ANOVA) for SOX2 expression. (b, c) Fluorescence Intensity of PRL3 (cytoplasmic) and OCT4 (nuclear) were analyzed in CHO-PRL3-PGCC cells, CHO-PRL3 cells and CHO cells of same images. (b) Both PRL3 & OCT4 expression were highest in CHO-PRL3-PGCC cells indicating high PRL3 expression is associated with high OCT4 expression. (c) Fluorescence intensity of OCT4 from (b). $P < 0.0001$ (Unpaired t test) for PRL3 expression. $P < 0.0001$ (One-way ANOVA) for OCT4 expression. Data represent mean \pm SD from 2 biological replicates.

Supplementary Figure 3. gH2AX is highly expressed in CHO-PRL3- PGCC cells.

Fluorescence Intensity of PRL3 (cytoplasmic) and gH2AX (nuclear) were analyzed in CHO-PRL3-PGCC cells, CHO-PRL3 cells and CHO cells of same images. (a) Both PRL3 & gH2AX expression were highest in CHO-PRL3-PGCC cells indicating high PRL3 expression is associated with high gH2AX expression. (b) Fluorescence intensity of OCT4 from (a). $P < 0.0001$ (Unpaired t test) for PRL3 expression. $P < 0.0001$ (One-way ANOVA) for gH2AX expression. Data represent mean \pm SD from 3 biological replicates.

References:

1. Zeng Q, Dong JM, Guo K, Li J, Tan HX, Koh V, PRL-3 and PRL-1 promote cell migration, invasion, and metastasis. *Cancer Res.* **63**, 2716-22 (2003).
2. Guo K, Tang JP, Li J, Zeng Q, Monoclonal antibodies target intracellular PRL phosphatases to inhibit cancer metastases in mice. *Cancer Biology & Therapy.* **7(5)**, 750-757 (2008).

Reviewers' Comments:

Reviewer #1:

Remarks to the Author:

The authors addressed all this reviewer's concerns.

Reviewer #2:

Remarks to the Author:

The authors have appropriately addressed my concerns. Congratulations for a job well done!

Response to the reviewer's comments

Reviewer #1 (Remarks to the Author):

The authors addressed all this reviewer's concerns.

Authors: We thank the reviewer for his kind understanding and support to our research.

Reviewer #2 (Remarks to the Author):

The authors have appropriately addressed my concerns. Congratulations for a job well done!

Authors: We appreciate the reviewer's strong support and compliment on our research.